# Causal Discovery in Semi-Stationary Time Series

**Shanyun Gao**
Purdue University
gao565@purdue.edu

**Raghavendra Addanki**
Adobe Research
raddanki@adobe.com

**Tong Yu**
Adobe Research
tyu@adobe.com

**Ryan A. Rossi**
Adobe Research
ryrossi@adobe.com

**Murat Kocaoglu**
Purdue University
mkocaoglu@purdue.edu

## Abstract

Discovering causal relations from observational time series without making the stationary assumption is a significant challenge. In practice, this challenge is common in many areas, such as retail sales, transportation systems, and medical science. Here, we consider this problem for a class of non-stationary time series. The structural causal model (SCM) of this type of time series, called the *semi-stationary* time series, exhibits that a finite number of different causal mechanisms occur sequentially and periodically across time. This model holds considerable practical utility because it can represent periodicity, including common occurrences such as seasonality and diurnal variation. We propose a constraint-based, non-parametric algorithm for discovering causal relations in this setting. The resulting algorithm, PCMCI$_\Omega$, can capture the alternating and recurring changes in the causal mechanisms and then identify the underlying causal graph with conditional independence (CI) tests. We show that this algorithm is sound in identifying causal relations on discrete-valued time series. We validate the algorithm with extensive experiments on continuous and discrete simulated data. We also apply our algorithm to a real-world climate dataset.

## 1 Introduction

In modern sciences, causal discovery aims to identify the collection of causal relations from observational data, as in Pearl [1980], Peters et al. [2017] and Spirtes et al. [2000]. One of the most popular causal discovery approaches is the so-called constraint-based method. Constraint-based approaches assume that the probability distribution of variables is causal Markov and faithful to a directed acyclic graph called the causal graph. Given large enough data, they can then recover the corresponding Markov equivalence class by exploiting conditional independence relationships of the variables. See Peters et al. [2017]. There are many constraint-based algorithms such as PC and FCI algorithms Spirtes et al. [2000]. The standard assumption of these approaches is that data samples are independent and identically distributed, which makes it possible to perform CI tests. See Bergsma [2004], Zhang et al. [2012] and Shah and Peters [2020].

Recently, there have been numerous efforts to extend such constraint-based algorithms to accommodate time series data. For instance, PCMCI in Runge et al. [2019] and LPCMCI in Gerhardus and Runge [2020] are the PC-based algorithms for time series. Inspired by FCI algorithms, approaches designed for time series include ANLSTM in Chu and Glymour [2008], tsFCI in Entner and Hoyer [2010] and SVAR-FCI in Malinsky and Spirtes [2018]. This setup is relevant in several industrial applications since many data points have an associated time-point, such as root-cause analysis in Ikram et al. [2022]. Most of the existing causal discovery algorithms make the stationary assumption.

37th Conference on Neural Information Processing Systems (NeurIPS 2023).

See Chu and Glymour [2008], Hyvärinen et al. [2010], Entner and Hoyer [2010], Peters et al. [2013], Malinsky and Spirtes [2018], Runge et al. [2019], Pamfil et al. [2020]and Assaad et al. [2022].

Non-stationary temporal data makes causal discovery more challenging since the statistics are time-variant, and it is unreasonable to expect that the underlying causal structure is time-invariant. Identifying causal relations from non-stationary time series without imposing any restriction on the data is difficult. Here, we focus on a specific class of non-stationary time series, called the *semi-stationary* time series, whose structural causal model (SCM) exhibits that a finite number of different causal mechanisms occur sequentially and periodically across time. One example is illustrated in Fig. 1, where the time series $\mathbf{X}^1$ has three different causal mechanisms across time, shown as red edges, green edges, and blue edges, respectively. Similarly, time series $\mathbf{X}^2$ has two alternative causal mechanisms. This setting holds considerable practical utility. Periodic nature is commonly observed in many real-world time series data. See Han et al. [2002], Nakamura et al. [2003], Carskadon et al. [2005] and Komarzynski et al. [2018]. Here are a few additional intuitive examples: poor traffic conditions often coincide with commute time and weekends; household electricity consumption typically follows a pattern of being higher at night and lower during the daytime. Consequently, it is reasonable to expect periodic changes in the causal relations underlying this type of time series without assuming stationarity. Here, the constraint-based methods in Chu and Glymour [2008], Entner and Hoyer [2010], Malinsky and Spirtes [2018] and Runge et al. [2019], designed for stationary time series, may fail. Given observational data with periodically changing causal structures, it is hard to apply CI tests directly. Most of the other algorithms designed for non-stationary time series rely heavily on model assumptions, as in Gong et al. [2015], Pamfil et al. [2020], and Huang et al. [2019]. These algorithms are discussed further in the related work.

In this paper, we propose an algorithm to address this problem, namely non-parametric causal discovery in time series data with semi-stationary SCMs. The key contributions of our work are:

- We develop an algorithm to discover the causal structure from semi-stationary time series data where the underlying causal structures change periodically. Our algorithm systematically uses the PCMCI algorithm proposed for the stationary setting in Runge et al. [2019]. The resulting algorithm is hence named PCMCI$_\Omega$ where $\Omega$ denotes periodicity.

- We validate our method with synthetic simulations on both continuous-valued and discrete-valued time series, showing that our method can correctly learn the periodicity and causal mechanism of the synthetic time series.

- We utilize our method in a real-world climate application. The result reveals the potential existence of periodicity in those time series, and the stationary assumption made by previous works could be relaxed in some practical situations.

## 1.1 Related Work

PCMCI has been applied in diverse domains to investigate atmospheric interactions in the biosphere, as demonstrated in Krich et al. [2020], global wildfires as explored in Qu et al. [2021], water usage as studied in Zou et al. [2022], ultra-processed food manufacturing as examined in Menegozzo et al. [2020], and causal feature selection as discussed in Peterson [2022], among other applications. See Arvind et al. [2021], Gerhardus and Runge [2020], Castri et al. [2023a,b]. While PCMCI has achieved considerable success, it is not without its limitations. One notable assumption that can be challenged is the concept of *causal stationary*, that is, causal relations are time-invariant. PCMCI exhibits robustness when applied to linear models with an added non-stationary trend. See also Runge et al. [2019]. However, there is an ongoing exploration to enhance its performance in a wider range of non-stationary settings.

Although not as extensively as the stationary case, causal discovery in non-stationary time series has been studied by some authors. However, many of those algorithms rely on parametric assumptions such as the vector autoregressive model in Gong et al. [2015] and Malinsky and Spirtes [2019]; linear and nonlinear state-space model in Huang et al. [2019]. One non-parametric algorithm in the literature is CD-NOD proposed by Huang et al. [2020], which has been extended to recover time-varying instantaneous and lagged causal relationships. In very recent work, Fujiwara et al. [2023] proposed an algorithm JIT-LiNGAM to obtain a local approximated linear causal model combining algorithm LiNGAM and JIT framework for non-linear and non-stationary data. To the best of our knowledge, no other non-parametric approaches can discover causal relations underlying time series without

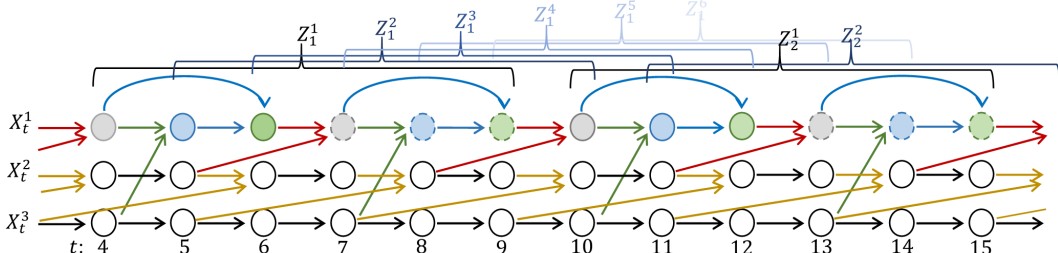

Figure 1: Partial causal graph for 3-variate time series $V = \{\mathbf{X}^1, \mathbf{X}^2, \mathbf{X}^3\}$ with a Semi-Stationary SCM where $\tau_{\max} = 3$, $\omega_1 = 3$, $\omega_2 = 2$, $\omega_3 = 1$, $\Omega = 6$ and $\delta = 6$. The first $3(=\tau_{\max})$ time slices $\{\mathbf{X}_t\}_{1 \leq t \leq 3}$ are the starting points. The same color edges represent the same causal mechanism. E.g. for $\mathbf{X}^1$: there are 3 ($= \omega_1$) time partition subsets $\{\Pi_k^1\}_{1 \leq k \leq 3}$. The time points $t$ of nodes $X_t^1$ sharing the same filling color are in the same time partition subsets. The time points $t$ of nodes $X_t^1$ sharing both the same filling color and the same outline shape are in the same homogenous time partition subsets (the definitions are in the supplementary material). There are 6 ($= \delta$) different Markov chains in this multivariate time series $V$, and the first element of these 6 Markov chains is shown as $\{Z_1^q\}_{1 \leq q \leq 6}$ and are tinted with a gradient of blue hues. The superscript $q$ of $Z_i^q$ is the index of different Markov chains, whereas the subscript $i$ denotes the running index of that specific Markov chain. For instance, $Z_1^1$ and $Z_2^1$ denote the first two elements of the first Markov chain, while $Z_1^2$ and $Z_2^2$ denote the first two elements of the second Markov chain.

assuming stationarity and can also allow for sudden changes in causal mechanisms. Our proposed approach does not directly enforce the stationary assumption on the time series. The SCM also integrates a finite set of causal mechanisms that exhibit periodic variations over time.

## 2 PCMCI$_\Omega$: Capturing Periodicity of the Causal Structure

In this section, we formulate the problem of learning the causal graph on multi-variate time series data when the SCM exhibits periodicity in causal mechanisms. In section 2.1, we present the necessary definitions and provide an overview of the problem setting. In section 2.2, we introduce the required assumption.

### 2.1 Preliminaries

Let $\mathcal{G}(V, E)$ denote the underlying causal graph, and for each variable $X \in V$, we denote the set of all incoming neighbors as parents, denoted by $\mathrm{Pa}(X)$.

For any two variables $X, Y \in V$ and $S \subset V$, we denote the CI relation $X$ is independent of $Y$ conditioned on $S$, by $X \perp\!\!\!\perp Y \mid S$.

For simplicity's sake, define sets: $[b] := \{1, 2, ..., b\}$ and $[a, b] := \{a, a+1, ..., b\}$, where $a, b \in \mathbb{N}$.

In the time series setting, let $X_t^j \in \mathbb{R}^1$ denote the variable of $j$th time series at time $t$, $\mathbf{X}^j = \{X_t^j\}_{t \in [T]} \in \mathbb{R}^T$ denote a univariate time series and $\mathbf{X}_t = \{X_t^j\}_{j \in [n]} \in \mathbb{R}^n$ denote a slice of all variables at time point $t$. $V = \{\mathbf{X}^j\}_{j \in [n]} = \{\mathbf{X}_t\}_{t \in [T]} \in \mathbb{R}^{n \times T}$ denotes a $n$-variate time series. By default, we assume $n > 1$ and hence $\mathbf{X}^j \subsetneq V$, and $p(V) \neq 0$, where $p(.)$ denotes the probability or probability density.

**Definition 2.1** (*Non-Stationary* SCM). A Non-Stationary Structural Causal Model (SCM) is a tuple $\mathcal{M} = \langle V, \mathcal{F}, \mathcal{E}, \mathbb{P} \rangle$ where there exists a $\tau_{\max} \in \mathbb{N}^+$, defined as:

$$\tau_{\max} := \arg\max_\tau \{\tau : X_{t-\tau}^i \in \mathrm{Pa}(X_t^j), i, j \in [n]\}, \tag{1}$$

such that with this $\tau_{\max}$, each variable $X_{t > \tau_{\max}}^j \in V$ is a deterministic function of its parent set $\mathrm{Pa}(X_{t > \tau_{\max}}^j) \in V$ and an unobserved (exogenous) variable $\epsilon_{t > \tau_{\max}}^j \in \mathcal{E}$:

$$X_t^j = f_{j,t}(\mathrm{Pa}(X_t^j), \epsilon_t^j), \; j \in [n], t \in [\tau_{\max} + 1, T], \tag{2}$$

and there exist at least two different time points $t_0, t_1 \in [\tau_{\max} + 1, T]$ satisfying

$$f_{j,t_0} \neq f_{j,t_1}, \ \exists j \in [n], \{t_0, t_1\} \subset [\tau_{\max} + 1, T]. \tag{3}$$

where $f_{j,t}, f_{j,t_0}, f_{j,t_1} \in \mathcal{F}$ and $\{\epsilon_t^j\}_{t \in [T]}$ are jointly independent with probability measure $\mathbb{P}$. $\tau_{\max}$ is the finite maximal lag in terms of the causal graph $\mathcal{G}$.

**Definition 2.2** (*Semi-Stationary* SCM)**.** A Semi-Stationary SCM is a Non-Stationary SCM that additionally satisfies the following conditions. For each $j \in [n]$, there exists an $\omega \in \mathbb{N}^+$ such that:

$$a) \ f_{j,t} = f_{j,t+N\omega}, \tag{4}$$

$$b) \ \mathrm{Pa}(X_{t+N\omega}^j) = \{X_{s+N\omega}^i : X_s^i \in \mathrm{Pa}(X_t^j), i \in [n]\}, \tag{5}$$

$$c) \ \epsilon_t^j, \epsilon_{t+N\omega}^j \text{ are i.i.d.} \tag{6}$$

are satisfied for all $t \in [\tau_{\max} + 1, T], N \in \{N : N \in \mathbb{N}, t + N\omega \leq T\}$. This means that a finite number of causal mechanisms are repeated periodically for every univariate time series $\mathbf{X}^j$ in $V$. One example of this model is illustrated in Fig. 1. For $\mathbf{X}^1$ in Fig. 1, three causal mechanisms are reiterated periodically with $\omega_1 = 3$, represented by red, green, and blue edges, respectively.

The minimum value that satisfies the above conditions for $\mathbf{X}^j$ is defined as the *periodicity* of $\mathbf{X}^j$, denoted by $\omega_j$. Furthermore, for an $n$-variate time series $V$, $\Omega$ denotes the minimum periodicity across all time series $\mathbf{X}^j, j \in [n]$. The number of causal mechanisms occurring sequentially and periodically of univariate time series $\mathbf{X}^j$ is $\omega_j$, and that number of causal mechanisms of multivariate time series $V$ is $\Omega$. For $\mathbf{X}^j$, the causal mechanisms are associated with each variable $X_t^j$. However, for $V$, the causal mechanisms are related to each time slice vector $\mathbf{X}_t$ as a whole. The relationship between $\Omega$ and $\omega_j$ can be captured by:

$$\Omega = \mathrm{LCM}(\{\omega_j : j \in [n]\}) \tag{7}$$

where LCM(.) is an operation to find the least common multiple between any two or more numbers. Here, $\Omega$ is the smallest common multiple among $\{\omega_1, ... \omega_n\}$. In Fig. 1, $\Omega = \mathrm{LCM}(3, 2, 1) = 6$.

**Definition 2.3** (*Time Partition*)**.** A time partition $\Pi^j(T)$ of a univariate time series $\mathbf{X}^j$ in a Semi-Stationary SCM with periodicity $\omega_j$ is a way of dividing all time points $t \in [T]$ into a collection of non-overlapping non-empty subsets $\{\Pi_k^j(T)\}_{k \in [\omega_j]}$ such that:

$$\Pi_k^j(T) := \{t : \tau_{\max} + 1 \leq t \leq T, (t \bmod \omega_j) + 1 = k\}. \tag{8}$$

where $\bmod$ denotes the modulo operation. For instance, $5 \bmod 3 = 2$.

We can observe that the variables in $\{X_t^j\}_{t \in \Pi_k^j(T)}$ share the same causal mechanism. Since the number of potentially different causal mechanisms of variables in $\mathbf{X}^j$ is $\omega_j$, the number of such time partition subsets is $\omega_j$. For simplicity, notations $\Pi^j$ and $\Pi_k^j$ are used instead of $\Pi^j(T)$ and $\Pi_k^j(T)$. In Fig. 1, $\Pi_1^1 = \{4, 7, 10, 13, .., 4 + 3N, ...\}$, $\Pi_2^1 = \{5, 8, 11, 14, .., 5 + 3N, ...\}$ and $\Pi_3^1 = \{6, 9, 12, 15, .., 6 + 3N, ...\}$ where $N \in \mathbb{N}^+$. The nodes $X_t^1$ are classified into their associated time partition subsets by the matching colors.

**Definition 2.4** (*Illusory Parent Sets*)**.** For a univariate time series $\mathbf{X}^j \in V$ with Semi-Stationary SCM having periodicity $\omega_j > 1$, parent set index $\mathrm{pInd}_{k \in [\omega_j]}^j$ is defined as:

$$\mathrm{pInd}_k^j := \{(y_i, \tau_i)\}_{i \in [n']}, \text{ given } \mathrm{Pa}(X_t^j) = \{X_{t-\tau_1}^{y_1}, X_{t-\tau_2}^{y_2}, ..., X_{t-\tau_{n'}}^{y_{n'}}\}, \ \forall t \in \Pi_k^j \tag{9}$$

where $n' = |\mathrm{Pa}(X_t^j))|$, $\tau_i$ is the time lag and $y_i$ is the variable index. Given $\mathrm{pInd}_k^j$, *Illusory Parent Sets* are defined as:

$$\mathrm{Pa}_k(X_t^j) = \{X_{t-\tau_i}^{y_i} : (y_i, \tau_i) \in \mathrm{pInd}_k^j\}, \ \forall k \in \{k : t \notin \Pi_k^j\} \tag{10}$$

Put simply, the illusory parent sets of $X_t^j$ are the time-shift version of the parent set of other variables in $\mathbf{X}^j$ that have a different causal mechanism from $X_t^j$. Note that the illusory parent sets are constructed specifically in the Semi-Stationary SCM. For stationary SCM, there is no illusory parent set needed. To maintain consistency in notation, for time points $t \in \Pi_k^j$, the notation $\mathrm{Pa}_k(X_t^j)$ can also be extended to encompass the true parent set of $X_t^j$:

$$\mathrm{Pa}_k(X_t^j) := \mathrm{Pa}(X_t^j), \ \forall t \in \Pi_k^j \tag{11}$$

By doing so, $\mathrm{Pa}(X_t^j) \subset \cup_{k \in [\omega_j]} \mathrm{Pa}_k(X_t^j)$. In Fig. 1, by observing $\mathrm{Pa}(X_7^1)$, we have $\mathrm{pInd}_1^1 = \{(1,1),(2,2)\}$; by observing $\mathrm{Pa}(X_8^1)$, we have $\mathrm{pInd}_2^1 = \{(1,1),(3,1)\}$ and finally by observing $\mathrm{Pa}(X_9^1)$, $\mathrm{pInd}_3^1 = \{(1,1),(1,2)\}$. Based on those indexes, $Pa_1(X_7^1) = \{X_{7-1}^1, X_{7-2}^2\} = \{X_6^1, X_5^2\}$, $Pa_2(X_7^1) = \{X_{7-1}^1, X_{7-1}^3\} = \{X_6^1, X_6^3\}$ and $Pa_3(X_7^1) = \{X_{7-1}^1, X_{7-2}^1\} = \{X_6^1, X_5^1\}$. The first one is the true parent set of $X_7^1$ and the latter two are the illusory parent sets. The order of those parent sets is not important.

At last, we need to further define a series of Markov chains that are associated tightly with *Semi-Stationary* SCM. The presence and characteristics of these Markov chains are thoroughly examined in the supplementary materials. The motivation behind creating such Markov chains is to introduce assumptions on them rather than directly on the original data $V$.

**Definition 2.5** (*Time Series as a Markov Chain*). For time series $V$ with *Semi-Stationary* SCM, there are (potentially) $\delta$ different Markov chains $\{Z_n^q\}_{n \in \mathcal{N}}, q \in [\delta]$:

$$Z_n^q = \{\mathbf{X}_{\tau_{\max}+q+(n-1)\delta}, \mathbf{X}_{\tau_{\max}+q+1+(n-1)\delta}, ..., \mathbf{X}_{\tau_{\max}+q-1+n\delta}\},$$

where $\mathcal{N} := \{n \in \mathbb{N}^+ : \tau_{\max} + q - 1 + n\delta \leq T\}$, $\delta = \lceil \frac{\tau_{\max}+1}{\Omega} \rceil \Omega$. Note that in $\{Z_n^q\}$, $q$ is used to indicate a specific Markov chain, while $n$ serves as the running index for that particular Markov chain. Such a Markov chain $\{Z_n^q\}, q \in [\delta]$ exists as long as $\mathrm{Pa}(Z_n^q) \subset Z_n^q \cup Z_{n-1}^q$ for all $n$. This is a finite state Markov Chain if all time series in $V$ are discrete-valued time series. The state space of $\{Z_n^q\}$ is the set containing all possible realization of $\{\mathbf{X}_{\tau_{\max}+q+(i-1)+(n-1)\delta}\}_{i \in [\delta], n \in \mathbb{N}}$. The transition probabilities between the states are the product of associated causal mechanisms based on Assumption **A2**, which is elaborated by an example in section C.1 (Eq.(10)-(11)) of the supplementary material.

## 2.2 Assumptions for PCMCI$_\Omega$

**A1. Sufficiency**: There are no unobserved confounders.

**A2. Causal Markov Condition**: Each variable $X$ is independent of all its non-descendants, given its parents $\mathrm{Pa}(X)$ in $\mathcal{G}$.

**A3. Faithfulness Condition (Pearl [1980])**: Let $P$ be a probability distribution generated by $\mathcal{G}$. $\langle \mathcal{G}, P \rangle$ satisfies the Faithfulness Condition if and only if every conditional independence relation true in $P$ is entailed by the Causal Markov Condition applied to $\mathcal{G}$.

**A4. No Contemporaneous Causal Effects**: Edges between variables at the same time are not allowed.

**A5. Temporal Priority**: Causal relations that point from the future to the past are not permitted.

**A6. Hard Mechanism Change**: If at time points $t_1$ and $t_2$, the causal mechanisms of $X_{t_1}^j$ and $X_{t_2}^j$ are different, then their corresponding parent sets can not be transformed to each other by time shifts:

$$f_{j,t_1} \neq f_{j,t_2} \Rightarrow \mathrm{Pa}(X_{t_2}^j) \neq \{X_{s+(t_2-t_1)}^i : X_s^i \in \mathrm{Pa}(X_{t_1}^j), i \in [n]\}.$$

**A7. Irreducible and Aperiodic Markov Chain**: The Markov chains $\{Z_n\}$ of $V$ are assumed to be irreducible (Serfozo [2009]): for all states $i$ and $j$ of $\{Z_n\}$, $\exists n$ so that

$$p_{ij}^{(n)} := p(Z_{n+1} = j | Z_1 = i) > 0 \tag{12}$$

and aperiodic(Karlin [2014]): for every state $i$ of $\{Z_n\}$, $d(i) = 1$, where the period $d(i)$ of the state $i$ is the greatest common divisor of all integers $n$ for which $p_{ii}^{(n)} > 0$.

Assumptions **A1-A5** are conventional and commonly employed in causal discovery methods for time series data. On the other hand, our approach requires additional Assumptions **A6-A7** to be in place. To clarify, **A6** is essential because our method may encounter challenges in distinguishing distinct causal mechanisms for variables in $\{X_n^j\}_{n \in [T]}$ if they share identical parent sets after time shifts. As for **A7**, it serves a crucial role in establishing the soundness of our algorithm.

## 3 PCMCI$_\Omega$ Algorithm

In this section, we propose an algorithm called PCMCI$_\Omega$, and in section 3.1, we present a theorem demonstrating the soundness of PCMCI$_\Omega$ and its ability to recover the causal graph. Our algorithm

---

**Algorithm 1** PCMCI$_\Omega$

---

1: **Input:** A $n$-variate time series $V = (\mathbf{X}^1, \mathbf{X}^2, \mathbf{X}^3, ..., \mathbf{X}^n)$, periodicity upper bound $\omega_{\text{ub}}$, time lag upper bound $\tau_{\text{ub}}$. By default, we assume $\tau_{\text{ub}}$ and $\omega_{\text{ub}}$ are larger than their true value.

2: A superset of parent set is obtained using PCMCI with $\tau_{\text{ub}}$ and denote it by $\widehat{S\text{Pa}}(X_t^j)\ \forall j, t$.

3: **for** $\mathbf{X}^j$ where $j \in [n]$ **do**

4:     **for** a guess $\omega \in [\omega_{ub}]$ of $\omega_j$ **do**

5:         Let $\widehat{\Pi}^j := \{\widehat{\Pi}_k^j | k \in [\omega]\}$ where $\widehat{\Pi}_k^j = \{2\tau_{\text{ub}} + k, 2\tau_{\text{ub}} + \omega + k, 2\tau_{\text{ub}} + 2\omega + k, \cdots\}$.

6:         **for** $k \in [\omega]$ **do**

7:             Initialize the parent set for $X_t^j, t \in \{t : t \geq 2\tau_{\text{ub}}, t \in \widehat{\Pi}_k^j\}$ (with guess $\omega$) denoted by $\widehat{\text{Pa}}_\omega(X_t^j) \leftarrow \widehat{S\text{Pa}}(X_t^j)$.

8:             Consider $X_{t-\tau}^i \in \widehat{\text{Pa}}_\omega(X_t^j)$. Remove $X_{t-\tau}^i$ from $\widehat{\text{Pa}}_\omega(X_t^j)$ if $X_{t-\tau}^i \perp\!\!\!\perp X_t^j \mid \left(\widehat{S\text{Pa}}(X_t^j) \cup \widehat{S\text{Pa}}(X_{t-\tau}^i)\right) \setminus X_{t-\tau}^i$ using a CI Test with samples $t \in \{t : t \geq 2\tau_{\text{ub}}, t \in \widehat{\Pi}_k^j\}$.

9:             Store $\widehat{\text{Pa}}_\omega(X_t^j)$ for $X_t^j, t \in \{t : t \geq 2\tau_{\text{ub}}, t \in \widehat{\Pi}_k^j\}$.

10:         **end for**

11:     **end for**

12:     $\widehat{\omega}_j \leftarrow \arg\min_{\omega \in [\omega_{\text{ub}}]} \max_{k \in [\omega]} |\widehat{\text{Pa}}_\omega(X_{t \in \widehat{\Pi}_k^j}^j)|$.

13:     Set $\widehat{\text{Pa}}(X_t^j) \leftarrow \widehat{\text{Pa}}_{\hat{\omega}_j}(X_t^j)$ for $X_t^j, t \in \{t : t \geq 2\tau_{\text{ub}}\}$.

14: **end for**

15: **return** $\hat{\omega}_j$ and $\widehat{\text{Pa}}(X_t^j)\ \forall j \in [n], t \geq 2\tau_{\text{ub}}$.

---

PCMCI$_\Omega$ builds on the Algorithm PCMCI in Runge et al. [2019]. Additional details about PCMCI are provided in the supplementary material.

**Overview of Algorithm 1** PCMCI$_\Omega$. We assume that the periodicity and time lag are upper bounded by $\omega_{\text{ub}}$ and $\tau_{\text{ub}}$ respectively. Using PCMCI Runge et al. [2019], we obtain a superset of parents for every variable $X_t^j$ denoted by $\widehat{S\text{Pa}}(X_t^j)$ (line 2). Our goal is to identify the correct set of parents along with its periodicity for every variable in $V$. For a variable $X_t^j$, we guess its periodicity $\omega$ by iterating over all possible values in $[\omega_{\text{ub}}]$. Next, we construct time partition subsets $\widehat{\Pi}_k^j,\ k \in [\omega]$ based on the guess of periodicity $\omega$. In each time partition subset, we maintain a parent set, denoted by $\widehat{\text{Pa}}_\omega(X_t^j)$, initializing it with the superset $\widehat{S\text{Pa}}(X_t^j)$. Then we test the causal relations between $X_{t-\tau}^i \in \widehat{\text{Pa}}_\omega(X_t^j)$ and $X_t^j$ using a CI test on the sample $t \in \widehat{\Pi}_k^j$ (lines 6-10).

For each guess $\omega$, every variable in $\mathbf{X}^j$ should have its estimated parent set (line 9), and there are total $\omega$ potentially different parent set index $\text{pInd}_{k \in [\omega]}^j$ in $\mathbf{X}^j$. We return an estimate $\hat{\omega}_j$ that maximizes the sparsity of the causal graph (Lemma 3.4). Therefore, we select the value of $\omega \in [\omega_{\text{ub}}]$ that minimizes the maximum value of $|\widehat{\text{Pa}}_\omega(X_t^j)|,\ t \in [T]$ as the estimator of $\omega_j$ (line 12).

### 3.1 Theoretical Guarantees

Our main theorem shows that PCMCI$_\Omega$ recovers the true causal graph on discrete data. There are three important lemmas. We provide all the detailed proof in the supplementary material.

**Theorem 3.1.** *Let $\hat{\mathcal{G}}$ be the estimated graph using the Algorithm PCMCI$_\Omega$. Under assumptions A1-A7 and with an oracle (infinite sample size limit), we have that:*

$$\hat{\mathcal{G}} = \mathcal{G} \tag{13}$$

*almost surely.*

Lemma 3.2 and Lemma 3.3 jointly state that if CI tests are conducted on samples generated by different causal mechanisms, the obtained parent sets $\widehat{S\text{Pa}}(X_t^j)$ should be the superset of the union of the true and illusory parent sets $\cup_k \text{Pa}_k(X_t^j)$. That is, the estimated graph should be denser than the correct graph. The true parent set can then be obtained by directly testing the independent relations between the target variable $X_t^j$ and the variables in $\widehat{S\text{Pa}}(X_t^j)$, assuming a consistent CI test. Note that the CI tests in our algorithm are assumed to be consistent given i.i.d. samples. We do not assume

the consistency of CI tests with respect to semi-stationary data. Therefore, any CI tests that maintain consistency with i.i.d. samples can be seamlessly integrated into our algorithm.

**Lemma 3.2.** *Denote that* $\{\mathrm{Pa}_k(X_t^j)\}_{k\in[\omega_j]}$ *contain the true and illusory parent sets, where* $\omega_j$ *is the true periodicity of* $\mathbf{X}^j$. *For any random variable* $X_t^j$ *with large enough t, under assumptions **A1-A7** and with an oracle (infinite sample size limit), we have:*

$$p\bigg(p(X_t^j\,|\cup_{k=1}^{\omega_j}\mathrm{Pa}_k(X_t^j)) \neq p(X_t^j\,|\cup_{k=1}^{\omega_j}\mathrm{Pa}_k(X_t^j)\setminus y)\bigg) = 1, \ \forall y \in \cup_{k=1}^{\omega_j}\mathrm{Pa}_k(X_t^j) \qquad (14)$$

Here, $p(X_t^j\,|\cup_{k=1}^{\omega_j}\mathrm{Pa}_k(X_t^j)) = \lim_{T\to\infty}\hat{p}(X_t^j\,|\cup_{k=1}^{\omega_j}\mathrm{Pa}_k(X_t^j))$ where $\hat{p}(X_t^j\,|\cup_{k=1}^{\omega_j}\mathrm{Pa}_k(X_t^j))$ is an estimated conditional distribution using all samples $t\in[\tau_{\max}+1,T]$:

$$\hat{p}(X_t^j\,|\cup_{k=1}^{\omega_j}\mathrm{Pa}_k(X_t^j)) = \frac{\sum_t \mathbb{1}(X_t^j,\cup_{k=1}^{\omega_j}\mathrm{Pa}_k(X_t^j))}{\sum_t \mathbb{1}(\cup_{k=1}^{\omega_j}\mathrm{Pa}_k(X_t^j))}. \qquad (15)$$

*Proof sketch.* We argue that the estimated conditional distribution in Eq.(15) can be written as a linear combination of $\hat{p}(X_t^j|\mathrm{Pa}(X_t^j))$ where $t\in\Pi_k^j, k\in[\omega_j]$, i.e., as a mixture of conditional distributions. The coefficients in the linear function, say $\alpha_k, k\in[\omega_j]$, can be further decomposed based on a finer time partition called the *homogenous time partition*, which consists of subsets constructed according to the Markov chains $\{Z_n^q\}_{q\in[\delta]}$ corresponding to the time series. Based on Assumption **A7**, the Markov chains are stationary and ergodic. Therefore, after sufficiently large time steps, the distribution of $\{Z_n^q\}_{q\in[\delta]}$ will be invariant across $n$ as it achieves unique equilibrium. With this type of stationary sample, we can express $\alpha_k$ by joint distributions instead of the indicators. Then, we can complete the proof of our inequality claim in Eq.(14) using Assumption **A2** and Bayes theorem.

$\square$

**Lemma 3.3.** *Let* $\widehat{SPa}(\mathbf{X}_t^j)$ *denote the estimated superset of parent set for* $\mathbf{X}^j\in V$ *obtained from the Algorithm 1 (line 2).* $\{\mathrm{Pa}_k(X_t^j)\}_{k\in[\omega_j]}$ *contain the true and illusory parent sets, where* $\omega_j$ *is the true periodicity of* $\mathbf{X}^j$. *Under assumptions **A1-A7** and with an oracle (infinite sample size limit), we have that:*

$$\cup_{k=1}^{\omega_j}\mathrm{Pa}_k(X_t^j) \subseteq \widehat{SPa}(X_t^j), \ \forall t\in[\tau_{\max}+1,T]$$

*almost surely.*

*Proof sketch.* Assume the contrary, i.e., there exists $s\in\cup_k\mathrm{Pa}_k(X_t^j)\setminus\widehat{SPa}(X_t^j)$. From Lemma 3.2, we have $X_t^j\not\perp\!\!\!\perp s\,\big|\cup_{k=1}^{\omega_j}\mathrm{Pa}_k(X_t^j)\setminus s$. By the Definition 2.4, we have $\mathrm{Pa}(X_t^j)\subset\cup_{k=1}^{\omega_j}\mathrm{Pa}_k(X_t^j)$. If $s\notin\mathrm{Pa}(X_t^j)$, by the causal Markov property (Assumption **A2**), the dependence relation can not be true because $s$ is a non-descendant of $X_t^j$. If $s\in\mathrm{Pa}(X_t^j)$, our Algorithm would have concluded that $X_t^j\not\perp\!\!\!\perp s\,\big|\widehat{SPa}(X_t^j)$ (line 2), evident from the causal Markov property, contradicting our assumption. Hence, the lemma. $\square$

Based on Lemma 3.2 and Lemma 3.3, we can identify the true $\omega_j$ for $\mathbf{X}^j$ through Lemma 3.4.

**Lemma 3.4.** *Let* $\omega_j$ *denote the true periodicity for* $\mathbf{X}^j\in V$ *and* $\widehat{\mathrm{Pa}}_\omega(X_{t\in\Pi_k^j}^j)$ *denote the estimated parent set for* $X_t^j$ *obtained from Algorithm 1 line 9, where* $t\in\Pi_k^j$. *Define:*

$$\widehat{\omega}_j = \arg\min_{\omega\in[\omega_{ub}]}\max_{k\in[\omega]}|\widehat{\mathrm{Pa}}_\omega(X_{t\in\Pi_k^j}^j)| \qquad (16)$$

*Under assumptions **A1-A7** and with an oracle (infinite sample limit), we have that* $\hat{\omega}_j=\omega_j, \ \forall j\in[n]$ *almost surely.*

*Proof sketch.* Assume the contrary that $\hat{\omega}_j\neq\omega_j$, then in the Algorithm 1, we have an incorrect time partition $\widehat{\Pi}^j$. Hence, CI tests that are performed use samples with different causal mechanisms.

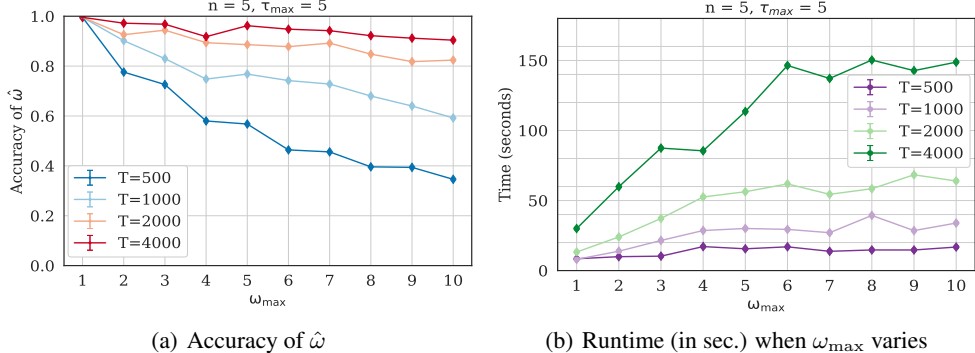

(a) Accuracy of $\hat{\omega}$         (b) Runtime (in sec.) when $\omega_{\max}$ varies

Figure 2: PCMCI$_\Omega$ is tested on 5-variate time series with $\tau_{\max} = 5$. Set $\tau_{\mathrm{ub}} = 15, \omega_{\mathrm{ub}} = 15$ for all variables. Every line corresponds to a different time series length. Every marker corresponds to the average accuracy rate or average running time over 100 trials. a) The accuracy rate of $\hat{\omega}$ for different time series lengths and different $\omega_{\max}$. b) Illustration of Runtime (in sec.) when $\omega_{\max}$ varies.

$\hat{p}(X_t^j | \cup_{k=1}^{\omega_j} \mathrm{Pa}_k(X_t^j))$ in Eq.(15) is estimated from a mixture of two or more time partition subsets, say $\Pi_1^j$ and $\Pi_2^j$. We can apply Lemma 3.2 with $\cup_{k=1}^2 \mathrm{Pa}_k(X_t^j)$. With Lemma 3.3, $\cup_{k=1}^2 \mathrm{Pa}_k(X_t^j) \subseteq \widehat{\mathrm{Pa}}_{\hat{\omega}_j}(X_t^j)$. Hence, for $\hat{\omega}_j$, $|\widehat{\mathrm{Pa}}_{\hat{\omega}_j}(X_t^j)| \geq |\cup_{k=1}^2 \mathrm{Pa}_k(X_t^j)|$ using mixture samples $t \in \cup_{k=1}^2 \Pi_k^j$. For true $\omega_j$, we have $|\widehat{\mathrm{Pa}}_{\omega_j}(X_t^j)| = |\mathrm{Pa}(X_t^j)|$. With Assumption **A6** the Hard Mechanism Change, $|\cup_{k=1}^2 \mathrm{Pa}_k(X_t^j)| > |\mathrm{Pa}(X_t^j)|$ so that $\omega_j$ always leads to a smaller size of estimated parent sets than $\hat{\omega}_j$, contrary to the definition of $\hat{\omega}_j$. Hence, $\hat{\omega}_j = \omega_j$. □

## 4 Experiments

### 4.1 Experiments on Continuous-valued Time Series

To validate the correctness and effectiveness of our algorithm, we perform a series of experiments. The Python code is provided at https://github.com/CausalML-Lab/PCMCI-Omega. In this section, we test four algorithms[1], PCMCI$_\Omega$, PCMCI Runge et al. [2019], VARLiNGAM Hyvärinen et al. [2010] and DYNOTEARS Pamfil et al. [2020], on continuous-valued time series with Gaussian noise. The experiments for continuous-valued time series with exponential noise and binary-valued time series are in the supplementary material.

Following Runge et al. [2019], we generate the continuous-valued time series in three steps:

1. Construct an $n$-variate time series $V$ with length $T$ using independent and identical (Standard Gaussian or Exponential) noise temporarily. Determine $\tau_{\max}$ and $\omega_{\max}$ where $\omega_{\max} = \max\{\omega_j\}_{j \in \{n\}}$. After making sure that one univariate time series, say $\mathbf{X}^j$, has periodicity $\omega_{\max}$, the periodicity of the remaining time series $\mathbf{X}^i, i \neq j$ is randomly selected from $\{1, \cdots, \omega_{\max}\}$ respectively.

2. Randomly generate $\omega_j$ binary edge matrices with shape $(n, \tau_{\max})$ for each time series $\mathbf{X}^j, j \in [n]$. 1 denotes an edge and 0 denotes no edge. Each binary matrix represent one parent set index $\mathrm{pInd}_k^j, k \in [\omega_j]$. Randomly generate $\omega_j$ coefficient matrices with shape $(n, \tau_{\max})$ for each time series $\mathbf{X}^j, j \in [n]$. One binary edge matrix and one coefficient matrix jointly determine one causal mechanism. Hence, total $\omega_j$ causal mechanisms are constructed. Here, make sure that $V$ satisfies Assumption **A6**.

3. Starting from time point $t > \tau_{\max}$, generate vector $\mathbf{X}_t$ over time according to all the causal mechanisms of $V$, until $t$ achieves $T$.

---

[1]We did not conduct experiments on JIT-LiNGAM because this is from a very recent paper Fujiwara et al. [2023] and is considered concurrent per NeurIPS policy.

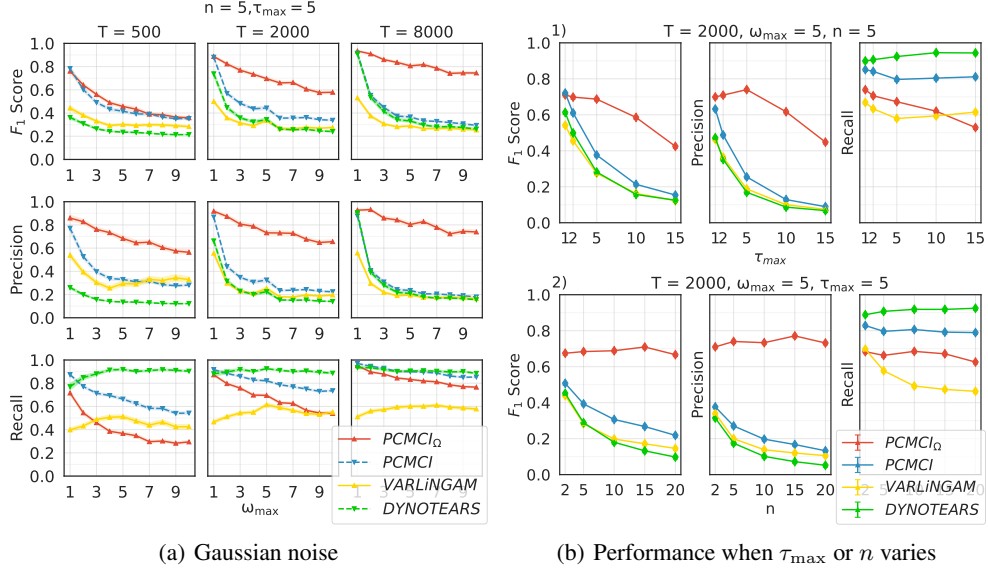

<table>
| (a) Gaussian noise | (b) Performance when $\tau_{\max}$ or $n$ varies |
</table>

Figure 3: 4 algorithms are tested on 5-variate time series. Set $\tau_{\text{ub}} = 15, \omega_{ub} = 15$ for all variables. Every line corresponds to a different algorithm. Every marker corresponds to the average performance over 100 trials.

Following the previous work in Huang et al. [2020], $F_1$ score, Adjacency Precision, and Adjacency Recall are used to measure the performance of the algorithms. The details of calculating these metrics are described in the Appendix. All the performance statistics are averaged over 100 trials. The standard error of the averaged statistics is displayed either by color filling or by error bars.

A correct estimator $\hat{\omega}$ is the prerequisite for obtaining the correct causal graph. Fig.2(a) shows the accuracy rate of $\hat{\omega}$ for different time lengths $T$. Here, elements in $\{N\omega_j\}$ where $N \in [\frac{\omega_{\text{ub}}}{\omega_j}]$ are all treated as correct estimations. By Definition 2.2 and 2.3, the multiple of $\omega_j$ is still associated with a correct causal graph. However, it leads to a finer time point partition $\Pi^j$, decreasing the sample size used in each CI test from approximately $T/\omega_j$ to approximately $T/(N\omega_j)$. The accuracy rate is sensitive to $\omega_{\max}$ for small $T$. This result verifies that algorithm PCMCI$_\Omega$ has the capacity to detect the true periodicity of each $\mathbf{X}^j \in V$ with a large enough time length.

We evaluate the performances of PCMCI$_\Omega$ on continuous-valued time series with Gaussian noise shown as Fig.3(a). As $T$ increases, it is natural to see a continuous improvement in performance. The sub-figures show that all three evaluation metrics decrease when $\omega_{\max}$ gets larger. The precision of PCMCI$_\Omega$ is always far better than other algorithms when $\omega_{\max}$ is not equal to 1. Given the fact that the parent sets $\widehat{\text{Pa}}(X_t^j) \, \forall j, t$ obtained from PCMCI$_\Omega$ are subsets of the parent set $\widehat{\text{SPa}}(X_t^j) \, \forall j, t$ estimated from PCMCI, the recall rate of PCMCI should be the upper bound of the recall rate of PCMCI$_\Omega$. This assertion has been verified as the red recall line of PCMCI$_\Omega$ is always below the blue recall line of PCMCI as $T$ increases.

In Fig.3(a), the recall of PCMCI$_\Omega$ is worse than PCMCI for $T = 500$. In this regime, the accuracy rate of $\hat{\omega}$ is low, shown as the dark blue line in Fig.2(a). Small sample sizes in CI tests may result in a sparser causal graph. Hence the number of true positive edges may decrease. This is a common problem for many constraint-based algorithms, but it hurts PCMCI$_\Omega$ the most because in PCMCI$_\Omega$, the sample sizes in each CI test are approximate $T/\hat{\omega}$ instead of $T$. As $T$ increases, the red recall line of PCMCI$_\Omega$ push forward to the blue recall line of PCMCI. The high value of both adjacent precision and recall rate with large $T$ verify that PCMCI$_\Omega$ can identify the correct causal graph.

We also observe the performance of our algorithm as $\tau_{\max}$ and $N$ varies in Fig.3(b). As the performance of PCMCI$_\Omega$ is consistent over $n$-variate time series with different $n$, large $\tau_{\max}$ may lead to a smaller precision and recall rate.

## 4.2 Case Study

Here, we construct an experiment with a real-world climate time series dataset. In Runge et al. [2019], the authors tested dependencies among monthly surface pressure anomalies in the West Pacific and surface air temperature anomalies in the Central Pacific, East Pacific, and tropical Atlantic from 1948 to 2012. Our application explores the causal relations among the monthly mean of the same set of variables from 1948-2022 with 900 months. Let $X_t^{\mathrm{wp}}$ denote the monthly mean of surface pressure in the West Pacific, $X_t^{\mathrm{cp}}$, $X_t^{\mathrm{ep}}$ and $X_t^{\mathrm{ta}}$ denote the monthly mean of air temperature in the Central Pacific, East Pacific, and tropical Atlantic, respectively.

The parent sets for each variable obtained from PCMCI$_\Omega$ algorithms are shown in Table 1. Sets of true and illusory parents of a variable at time $t$ are separated by curly braces. For instance, variable $X_t^{\mathrm{wp}}$ with $\hat{\omega}_{\mathrm{wp}} = 1$ means that the causal mechanism of the surface pressure in the West Pacific remains invariant over time with the estimated parent set $\{X_{t-1}^{\mathrm{wp}}, X_{t-2}^{\mathrm{wp}}, X_{t-1}^{\mathrm{ep}}, X_{t-1}^{\mathrm{ta}}\}$. Only time series $\mathbf{X}^{\mathrm{cp}}$ has three different parent sets, including one true parent set and two illusory parent sets, which appear periodically over time. The three parent sets of $X_t^{\mathrm{cp}}$ imply that the causal effect from the tropical Atlantic air temperature $X_{t-1}^{\mathrm{ta}}$ to the Central Pacific air temperature $X_t^{\mathrm{cp}}$ would disappear every quarter of a year. Note that we do not have a ground truth in this case, and we do not possess the necessary knowledge in this area, so the significance of these results is under-explored. More discussion about this application can be found in the supplementary materials.

Table 1: Climate application results estimated from PCMCI$_\Omega$.

| **PCMCI$_\omega$** | | |
|---|---|---|
| $X$ | $\hat{\omega}$ | $\{\widehat{\mathrm{Pa}}_k\}_{k \in [\hat{\omega}]}$: true and illusory parent sets |
| $X_t^{\mathrm{wp}}$ | 1 | $\{X_{t-1}^{\mathrm{wp}}, X_{t-2}^{\mathrm{wp}}, X_{t-1}^{\mathrm{ep}}, X_{t-1}^{\mathrm{ta}}\}$ |
| $X_t^{\mathrm{cp}}$ | 3 | $\{X_{t-1}^{\mathrm{cp}}\}$; $\{X_{t-1}^{\mathrm{cp}}, X_{t-2}^{\mathrm{cp}}, X_{t-1}^{\mathrm{ta}}\}$; $\{X_{t-1}^{\mathrm{cp}}, X_{t-1}^{\mathrm{ta}}\}$ |
| $X_t^{\mathrm{ep}}$ | 1 | $\{X_{t-1}^{\mathrm{ep}}, X_{t-1}^{\mathrm{ta}}, X_{t-2}^{\mathrm{ta}}, X_{t-1}^{\mathrm{cp}}\}$ |
| $X_t^{\mathrm{ta}}$ | 1 | $\{X_{t-1}^{\mathrm{ta}}, X_{t-1}^{\mathrm{wp}}\}$ |

## 5  Conclusions

In this paper, we propose a non-parametric, constraint-based causal discovery algorithm PCMCI$_\Omega$ designed for semi-stationary time-series data, in which a finite number of causal mechanisms are repeated periodically. We establish the soundness of our algorithm and assess its effectiveness on continuous-valued and discrete-valued time series data. The algorithm PCMCI$_\Omega$ has the capacity to reveal the existence of periodicity of causal mechanisms in real-world datasets.

## 6  Acknowledgements

This research has been supported in part by NSF CAREER 2239375 and Adobe Research. We wish to convey our heartfelt gratitude to the anonymous reviewers for their invaluable and constructive feedback, which significantly contributed to enhancing the quality of the manuscript.

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

# Appendix

## A PCMCI Algorithm

The PCMCI algorithm is proposed by Runge et al. [2019], aiming to detect time-lagged causal relations in a window causal graph. There are two stages of PCMCI: the condition-selection stage and the causal discovery stage. In the first stage, unnecessary edges are removed based on the conditional independencies from an initialized partially connected graph where Assumption **A4-A5** should be satisfied. In the second stage, Momentary Conditional Independence tests (MCI) are used to further remove the false positive edges caused by autocorrelations in time series data. More specifically, these two steps can be briefly formalized as follows:

- $PC_1$ in Algorithm A1: Condition selection stage. $PC_1$ is a variant of the skeleton-discovery part of the PC algorithm in a more robust version named stable-PC Le et al. [2016]. The goal in this stage is to obtain a superset of the parents $\widehat{\mathrm{Pa}}(X_t^j)$ for all variables $X_{t\in[\tau_{\max}+1,T]}^{j\in[n]} \in \mathbf{V}$. Initialize $\widehat{\mathrm{Pa}}(X_t^j) = \{X_{t-\tau}^i\}_{i\in[n],\tau\in[\tau_{\max}]}$. $\widehat{\mathrm{Pa}}(X_t^j)$ will remove $X_{t-\tau}^i$ if

$$X_{t-\tau}^i \perp\!\!\!\perp X_t^j \left| \widehat{\mathrm{Pa}}(X_t^j)\backslash\{X_{t-\tau}^i\}\right. \tag{1}$$

- MCI in Algorithm A2: Causal discovery stage. In this stage, do MCI tests for all variable pairs $(X_{t-\tau}^i, X_t^j)$ with $i,j \in [n]$ and time delays $\tau \in [\tau_{\max}]$:

$$MCI(X_{t-\tau}^i, X_t^j | \widehat{\mathrm{Pa}}(X_t^j)\backslash\{X_{t-\tau}^i\}, \widehat{\mathrm{Pa}}(X_{t-\tau}^i)) \tag{2}$$

where $\widehat{\mathrm{Pa}}(X_t^j)$ and $\widehat{\mathrm{Pa}}(X_{t-\tau}^i)$ are estimated from the $PC_1$ stage.

Note that $\tau_{\max}$ in this section is the same as $\tau_{\mathrm{ub}}$ in the main paper, serving as the upper bound for the time lag that exhibits causal effects. On the other hand, $\tau_{\max}$ in the main paper denotes the maximum time lag observed within the multivariate time series. Essentially, in the main paper, $\tau_{\mathrm{ub}}$ is a parameter that must be fed into the algorithm, and $\tau_{\max}$ is observed from the true causal graph. As a default, we assume $\tau_{\mathrm{ub}}$ is configured with a value greater than $\tau_{\max}$, ensuring that the algorithm uncovers the correct causal relations. See Fig. 1 for more detail.

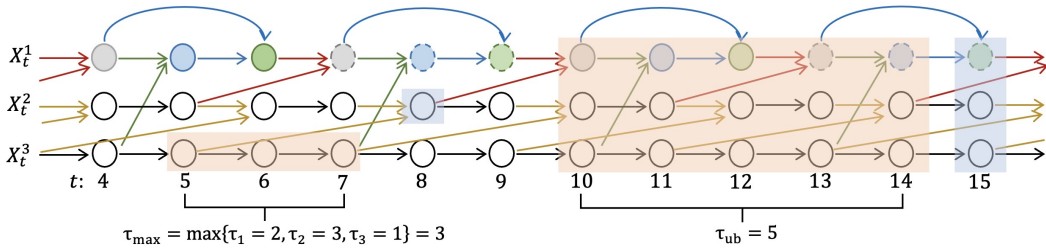

Figure 1: Set $\tau_{\mathrm{ub}}$ to be 5, then all parent candidates of variables at $t = 15$ are included in the large orange box, ranging from $t = 10$ to $t = 14$. Consequently, the algorithm will only examine causal effects with a time lag not exceeding 5. In the causal graph, $\tau_{\max}$ is 3, representing the maximum time lag observed among the 3-variate time series. Specifically, the maximum time lag for each component time series is $\tau_1 = 2, \tau_2 = 3, \tau_3 = 1$, respectively, and $\tau_{\max}$ represents the largest value among these three maximum lags.

---

**Algorithm A1** $PC_{q_{\max}}$

---

1: **Input:** A $n$-variate time series $V = (\mathbf{X}^1, \mathbf{X}^2, \mathbf{X}^3, ..., \mathbf{X}^n)$, target time series $\mathbf{X}^j$, maximum time lag $\tau_{\max}$, significance threshold $\alpha_{PC}$, maximum condition dimension $p_{\max}$ (default $p_{\max} = n\tau_{\max}$), maximum number of combinations $q_{\max}$ (default $q_{\max} = 1$), conditional independence test function $CI$

2: **function** $CI(X, Y, \mathbf{Z})$

3:     Test $X \perp\!\!\!\perp Y | \mathbf{Z}$ using test statistic measure $I$

4:     **return** $p$-value, test statistic value $I$

5: Initialize preliminary set of parents $\widehat{Pa}(X_t^j) = \{X_{t-\tau}^i : i \in \{1, ..., n\}, \tau \in \{1, ..., \tau_{\max}\}\}$

6: Initialize dictionary of test statistic values $I^{\min}(X_{t-\tau}^i \to X_t^j) = \infty \; \forall X_{t-\tau}^i \in \widehat{Pa}(X_t^j)$

7: **for** $p = 0, 1, 2, ..., p_{\max}$ **do**

8:     **if** $|\widehat{Pa}(X_t^j)| - 1 < p$ **then**

9:         Break for-loop

10:     **end if**

11:     **for all** $X_{t-\tau}^i$ in $\widehat{Pa}(X_t^j)$ **do**

12:         $q = -1$

13:         **for all** lexicographically chosen subsets $\mathcal{S} \subseteq \widehat{Pa}(X_t^j) \setminus \{X_{t-\tau}^i\}$ with $|\mathcal{S}| = p$ **do**

14:             $q = q + 1$

15:             **if** $q \geq q_{\max}$ **then**

16:                 Break from inner for-loop

17:             **end if**

18:             Run CI test to obtain $(p\text{-value}, I) \leftarrow CI(X_{t-\tau}^i, X_t^j, \mathcal{S})$

19:             **if** $|I| < I^{\min}(X_{t-\tau}^i \to X_t^j)$ **then**         ▷Store min. $I$ of parent among all tests

20:                 $I^{\min}(X_{t-\tau}^i \to X_t^j) = |I|$

21:             **end if**

22:             **if** $p\text{-value} > \alpha_{PC}$ **then**         ▷Removed only after all $X_{t-\tau}^i$ have been tested

23:                 Mark $X_{t-\tau}^i$ for removal from $\widehat{Pa}(X_t^j)$

24:                 Break from inner for-loop

25:             **end if**

26:         **end for**

27:     **end for**

28:     Remove non-significant parents from $\widehat{Pa}(X_t^j)$

29:     Sort parents in $\widehat{Pa}(X_t^j)$ by $I^{\min}(X_{t-\tau}^i \to X_t^j)$ from largest to smallest

30: **end for**

31: **return** $\widehat{Pa}(X_t^j)$

---

---

**Algorithm A2** $MCI$

---

1: **Input:** A $n$-variate time series $V = (\mathbf{X}^1, \mathbf{X}^2, \mathbf{X}^3, ..., \mathbf{X}^n)$, sorted parents $\widehat{Pa}(X_t^j)$ for all variables $X^j$ estimated with Algorithm A1, maximum time lag $\tau_{\max}$, maximum number $p_X$ of parents of variable $X^i$, and conditional independence test function $CI$

2: **for all** $(X_{t-\tau}^i, X_t^j)$ with $i, j \in \{1, ..., n\}, \tau \in \{0, ..., \tau_{\max}\}$, excluding $(X_t^j, X_t^j)$ **do**

3:     Remove $X_{t-\tau}^i$ from $\widehat{Pa}(X_t^j)$ if necessary

4:     Define $\widehat{Pa}_{p_X}(X_{t-\tau}^i)$ as the first $p_X$ parents from $\widehat{Pa}(X_t^i)$, shifted by $\tau$

5:     Run MCI test to obtain $(p\text{-value}, I) \leftarrow CI(X_{t-\tau}^i, X_t^j, \mathbf{Z} = \{\widehat{Pa}(X_t^j), \widehat{Pa}_{p_X}(X_{t-\tau}^i)\})$

6: **end for**

7: Optionally adjust $p$-value of all links by False Discovery Rate-approach (FDR)

8: **return** $p$-value and MCI test statistic values

---

# B  PCMCI$_\Omega$

For simplicity's sake, define sets: $[b] := \{1, 2, ..., b\}$ and $[a, b] := \{a, a+1, ..., b\}$, where $a, b \in \mathbb{N}$.

---

**Algorithm B1** $PCMCI_\Omega$

---

1: **Input:** A $n$-variate time series $V = (\mathbf{X}^1, \mathbf{X}^2, \mathbf{X}^3, ..., \mathbf{X}^n)$, periodicity upper bound $\omega_{\text{ub}}$, time lag upper bound $\tau_{\text{ub}}$. By default, we assume $\tau_{\text{ub}}$ and $\omega_{\text{ub}}$ are larger than their true value.

2: A superset of parent set is obtained using PCMCI with $\tau_{\text{ub}}$ and denote it by $\widehat{SPa}(X_t^j) \; \forall j, t$.

3: **for** $\mathbf{X}^j$ where $j \in [n]$ **do**

4:     **for** a guess $\omega \in [\omega_{ub}]$ of $\omega_j$ **do**

5:         Let $\widehat{\Pi}^j := \{\widehat{\Pi}_k^j | k \in [\omega]\}$ where $\widehat{\Pi}_k^j = \{2\tau_{\text{ub}} + k, 2\tau_{\text{ub}} + \omega + k, 2\tau_{\text{ub}} + 2\omega + k, \cdots \}$.

6:         **for** $k \in [\omega]$ **do**

7:             Initialize the parent set for $X_t^j, t \in \{t : t \geq 2\tau_{\text{ub}}, t \in \widehat{\Pi}_k^j\}$ (with guess $\omega$) denoted by $\widehat{\text{Pa}}_\omega(X_t^j) \leftarrow \widehat{SPa}(X_t^j)$.

8:             Consider $X_{t-\tau}^i \in \widehat{\text{Pa}}_\omega(X_t^j)$. Remove $X_{t-\tau}^i$ from $\widehat{\text{Pa}}_\omega(X_t^j)$ if $X_{t-\tau}^i \perp\!\!\!\perp X_t^j \mid \left( \widehat{SPa}(X_t^j) \cup \widehat{SPa}(X_{t-\tau}^i) \right) \setminus X_{t-\tau}^i$ using a CI Test with samples $t \in \{t : t \geq 2\tau_{\text{ub}}, t \in \widehat{\Pi}_k^j\}$.

9:             Store $\widehat{\text{Pa}}_\omega(X_t^j)$ for $X_t^j, t \in \{t : t \geq 2\tau_{\text{ub}}, t \in \widehat{\Pi}_k^j\}$.

10:         **end for**

11:     **end for**

12: ⋯⋯⋯⋯⋯⋯⋯⋯⋯⋯⋯⋯⋯⋯⋯⋯⋯⋯⋯⋯⋯⋯⋯⋯⋯⋯⋯⋯⋯⋯⋯⋯⋯⋯⋯

13:     **if** there exists *turning points* $S_j$, $S_j \in [\omega_{\text{ub}}]$ **then**

14:         $\widehat{\omega}_j \leftarrow \min S_j$

15:     **else**

16: ⋯⋯⋯⋯⋯⋯⋯⋯⋯⋯⋯⋯⋯⋯⋯⋯⋯⋯⋯⋯⋯⋯⋯⋯⋯⋯⋯⋯⋯⋯⋯⋯⋯⋯⋯

17:         $\widehat{\omega}_j \leftarrow \arg\min_{\omega \in [\omega_{\text{ub}}]} \max_{k \in [\omega]} |\widehat{\text{Pa}}_\omega(X_{t \in \widehat{\Pi}_k^j}^j)|$.

18:     **end if**

19:     Set $\widehat{\text{Pa}}(X_t^j) \leftarrow \widehat{\text{Pa}}_{\widehat{\omega}_j}(X_t^j)$ for $X_t^j, t \in \{t : t \geq 2\tau_{\text{ub}}\}$.

20: **end for**

21: **return** $\widehat{\omega}_j$ and $\widehat{\text{Pa}}(X_t^j) \; \forall j \in [n], t \geq 2\tau_{\text{ub}}$.

---

# C  Soundness of PCMCI$_\Omega$

## C.1  Stationary Markov Chain

**Claim**: Any discrete-valued time series $V$ with *Semi-Stationary* Structural Causal Model (SCM) satisfying assumption **A1, A2, A4, A5** can be written as a Markov chain $\{Z_n\}$ as long as this Markov chain satisfies $\text{Pa}(Z_n) \subset Z_n \cup Z_{n-1}$ for all $n$, where $Z_n$ is a set of variables in $V$. This Markov chain has a finite number of states if all time series in $V$ are discrete-valued time series.

Note that when the notation $n$ is related to a Markov chain $Z_n$, it means the running index. In the context of $X_t^{j \in [n]}$, $n$ represents the index of component time series within the $n$-variate time series.

To simplify, assume that one associated Markov chain of $V = \{\mathbf{X}, \mathbf{Y}\}$ has $Z_n = \{X_t, Y_t, X_{t-1}, Y_{t-1}\}$ with $t \in \{t \in \mathbb{N}^+ : t \leq T\}$ satisfying $\text{Pa}(Z_n) \subset Z_n \cup Z_{n-1}$. Here, the notation for the time points of variables is simplified as $t$ and $t-1$, even though it should be a function of $n$, the running index of the Markov chain. Note that $Z_{n-1} = \{X_{t-2}, Y_{t-2}, X_{t-3}, Y_{t-3}\}$ rather than $\{X_{t-1}, Y_{t-1}, X_{t-2}, Y_{t-2}\}$, as the simplified notation could erroneously suggest the latter sequence. A simple proof is shown below through Markov assumption (**A2**).

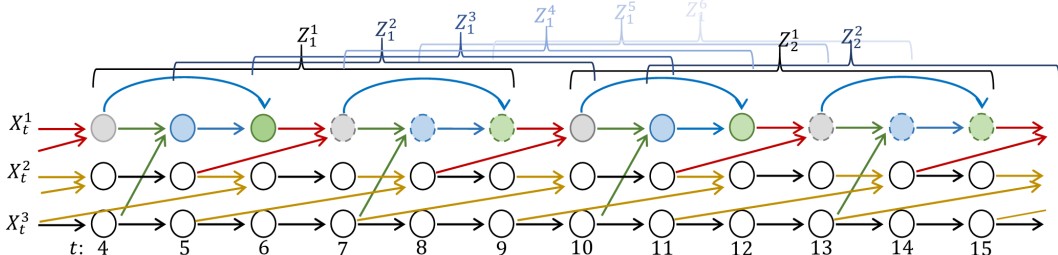

Figure 2: Partial causal graph for 3-variate time series $V = \{\mathbf{X}^1, \mathbf{X}^2, \mathbf{X}^3\}$ with a Semi-Stationary SCM where $\tau_{\max} = 3$, $\omega_1 = 3$, $\omega_2 = 2$, $\omega_3 = 1$, $\Omega = 6$ and $\delta = 6$. The first $3(=\tau_{\max})$ time slices $\{\mathbf{X}_t\}_{1 \le t \le 3}$ are the starting points. The same color edges denote the same causal mechanism. E.g. for $\mathbf{X}^1$: there are $3 (= \omega_j)$ time partition subsets $\{\Pi_k^1\}_{1 \le k \le 3}$. The time points $t$ of nodes $X_t^1$ sharing the same filling color are in the same time partition subsets. The time points $t$ of nodes $X_t^1$ sharing both the same filling color and the same outline shape are in the same homogenous time partition subsets. There are $6 (= \delta)$ different Markov chains in this multivariate time series $V$, and the first element of these 6 Markov chains is shown as $\{Z_1^q\}_{1 \le q \le 6}$ and are tinted with a gradient of blue hues. $Z_1^1$ and $Z_2^1$ denote the first two elements of the first Markov chain while $Z_1^2$ and $Z_2^2$ denote the first two elements of the second Markov chain.

*Proof.*

$$p(Z_n | Z_{n-1}, Z_{n-2}, ...) \tag{3}$$

$$= p(X_t, Y_t, X_{t-1}, Y_{t-1} | Z_{n-1}, Z_{n-2}, ...) \tag{4}$$

$$= p(X_t | Z_n \cup Z_{n-1} \setminus X_t, Z_{n-2}, \cdots) p\left(Y_t | Z_n \cup Z_{n-1} \setminus (X_t \cup Y_t), Z_{n-2}, \cdots\right) \cdots \tag{5}$$

$$= p\left(X_t | \mathrm{Pa}(X_t), Z_n \cup Z_{n-1} \setminus (X_t \cup \mathrm{Pa}(X_t))\right) \tag{6}$$

$$\times p\left(Y_t | \mathrm{Pa}(Y_t), Z_n \cup Z_{n-1} \setminus (X_t \cup Y_t \cup \mathrm{Pa}(Y_t))\right)$$

$$\times p\left(X_{t-1} | \mathrm{Pa}(X_{t-1}), Z_n \cup Z_{n-1} \setminus (X_t \cup Y_t \cup X_{t-1} \cup \mathrm{Pa}(X_{t-1}))\right) \cdots$$

$$= p(X_t | Z_n \cup Z_{n-1} \setminus X_t) p\left(Y_t | Z_n \cup Z_{n-1} \setminus (X_t \cup Y_t)\right) \cdots \tag{7}$$

$$= p(X_t, Y_t, X_{t-1}, Y_{t-1} | Z_{n-1}) \tag{8}$$

$$= p(Z_n | Z_{n-1}) \tag{9}$$

$\square$

Assume that the space of both $X_t$ and $Y_t$ with $t < T$ are $\{1, 2\}$. There are total $2^4 = 16$ states of Markov Chain $\{Z_n\} = \{\{X_t, Y_t, X_{t-1}, Y_{t-1}\}\}$. The transition probability $\mathbf{P}$ for this Markov Chain is illustrated as a $16 \times 16$ matrix:

$$\mathbf{P} = \begin{matrix} & \begin{matrix} (1,1,1,1) & (2,1,1,1) & \cdots \end{matrix} \\ \begin{matrix} (1,1,1,1) \\ (2,1,1,1) \\ \cdots \end{matrix} & \begin{bmatrix} p_{1,1} & p_{1,2} & \cdots \\ p_{2,1} & p_{2,2} & \cdots \\ \cdots & \cdots & \cdots \end{bmatrix}_{16 \times 16} \end{matrix}$$

where $(1, 1, 1, 1)$ means $X_t = 1, Y_t = 1, X_{t-1} = 1, Y_{t-1} = 1$. Each row in this transition probability matrix is a conditional distribution of $Z_n$ given one realization of $Z_{n-1}$. Each entry is a probability of having one specific realization of $Z_n$ given one realization of $Z_{n-1}$. This probability can be decomposed by conditional distributions based on Markov assumption (**A2**). Take $p_{1,1}$ as an example:

$$p_{1,1} = p(X_t = 1, Y_t = 1, X_{t-1} = 1, Y_{t-1} = 1 | X_{t-2} = 1, Y_{t-2} = 1, X_{t-3} = 1, Y_{t-3} = 1) \tag{10}$$

$$= p(X_t = 1 | \mathrm{Pa}(X_t)) p(Y_t = 1 | \mathrm{Pa}(Y_t)) p(X_{t-1} = 1 | \mathrm{Pa}(X_{t-1})) p(Y_{t-1} = 1 | \mathrm{Pa}(Y_{t-1})) \tag{11}$$

where $\mathrm{Pa}(.)$ here are realizations, not random variables.

For time series $V$ with *Semi-Stationary* SCM, there are (potentially) $\delta$ different Markov chains $\{Z_n^q\}, q \in [\delta]$:

$$Z_n^q = \{\mathbf{X}_{\tau_{\max}+q+(n-1)\delta}, \mathbf{X}_{\tau_{\max}+q+1+(n-1)\delta}, ..., \mathbf{X}_{\tau_{\max}+q-1+n\delta}\},$$

where $n \in \{n : n \in \mathbb{N}^+, \tau_{\max} + q - 1 + n\delta \leq T\}, \delta = \lceil \frac{\tau_{\max}+1}{\Omega} \rceil \Omega$. As proved in the claim, such a Markov chain exists as long as $\mathrm{Pa}(Z_n^q) \subset Z_n^q \cup Z_{n-1}^q$ for all $n$. The value of $\delta$ can guarantee the existence of such Markov chain because $\delta$ is larger than $\tau_{\max} + 1$ and is a multiple of $\Omega$, that is, a multiple of all $\{\omega_j\}_{j \in [n]}$. By doing so, $\mathrm{Pa}(Z_n^q) \subset Z_n^q \cup Z_{n-1}^q$ is satisfied; for any variable $X_t^j$, there exists $q \in [\delta]$ and $n \in \mathbb{N}^+$ such that variable $X_t^j$ and its parent set $\mathrm{Pa}(X_t^j)$ can be included in $Z_n^q$; and the causal mechanism generating $Z_n^q$ is invariant for different $n$. The state space of $\{Z_n^q\}$ is the set containing all possible realizations of $\{\mathbf{X}_{\tau_{\max}+q+(i-1)+(n-1)\delta}\}_{i \in [\delta], n \in \mathbb{N}}$. The transition probabilities between the states are the product of associated causal mechanisms based on Markov assumption (**A2**).

Determined by the starting slice $\mathbf{X}_t$ where $\tau_{\max} < t \leq \tau_{\max} + \delta$, there should be $\delta$ potentially different Markov chains $\{Z_n^q\}$ where $1 \leq q \leq \delta$. To be more specific, those Markov chains are:

$$\text{Markov Chain 1: } Z_n^1 = \{\mathbf{X}_{\tau_{\max}+1+(n-1)\delta}, \mathbf{X}_{\tau_{\max}+2+(n-1)\delta}, ..., \mathbf{X}_{\tau_{\max}+n\delta}\}, \tag{12}$$
$$\text{where } n \in \{n : n \in \mathbb{N}^+, \tau_{\max} + n\delta \leq T\}.$$

$$\text{Markov Chain 2: } Z_n^2 = \{\mathbf{X}_{\tau_{\max}+2+(n-1)\delta}, \mathbf{X}_{\tau_{\max}+3+(n-1)\delta}, ..., \mathbf{X}_{\tau_{\max}+1+n\delta}\}, \tag{13}$$
$$\text{where } n \in \{n : n \in \mathbb{N}^+, \tau_{\max} + 1 + n\delta \leq T\}.$$

$$\vdots$$

$$\text{Markov Chain } \delta: Z_n^\delta = \{\mathbf{X}_{\tau_{\max}+n\delta}, \mathbf{X}_{\tau_{\max}+1+n\delta}, ..., \mathbf{X}_{\tau_{\max}-1+(n+1)\delta}\}, \tag{14}$$
$$\text{where } n \in \{n : n \in \mathbb{N}^+, \tau_{\max} - 1 + (n+1)\delta \leq T\}.$$

Given Irreducible and Aperiodic Markov Chain assumption (**A7**), discrete-time Markov chain $\{Z_n^q\}_{0<n}, q \in [\delta]$ with finite states should be a stationary and ergodic Markov chain, and there is a unique stationary distribution $\pi_q$ (Bertsekas and Tsitsiklis [2008], Karlin [2014]). Additionally, the large power of the associate transition matrix $\mathbf{P}_q$ will eventually converge to a matrix in which each row is the stationary distribution $\pi_q$. Equivalently,

$$\lim_{n\to\infty} p(Z_n^q = a | Z_1^q = b) = p(Z_n^q = a), \forall a, b \in S. \tag{15}$$

where $S$ is the state space of $Z_n^q$.

In other words, after a sufficiently long time, equivalently, $n$ is large enough, the distribution of $\{Z_n^q\}$ does not change with increasing $n$. That is, for large enough $n$:

$$p(Z_{n_1}^q) = p(Z_{n_2}^q), \forall n_1, n_2 > n. \tag{16}$$

Returning from the stationary and ergodic Markov chains $\{Z_n^q\}, q \in [\delta]$ back to the original data $V$ through Eq.(12) to Eq.(14), the distribution of the original data $V$ must adhere to the following condition:

$$p(\mathbf{X}_{\tau_{\max}+q+n_1\delta}, \mathbf{X}_{\tau_{\max}+q+1+n_1\delta}, ..., \mathbf{X}_{\tau_{\max}+q+\delta-1+n_1\delta})$$
$$= p(\mathbf{X}_{\tau_{\max}+q+n_2\delta}, \mathbf{X}_{\tau_{\max}+q+1+n_2\delta}, ..., \mathbf{X}_{\tau_{\max}+q+\delta-1+n_2\delta}) \tag{17}$$

for any $q \in [\delta]$ and $n_1, n_2 > n$.

Given these clarifications, we can naturally introduce a more refined time partition that is based on, yet finer than, the time partition defined in Definition 2.3 in the main paper.

**Definition C.1** (*Homogenous Time Partition*). For a univariate time series $\mathbf{X}^j$ in a Semi-Stationary SCM with periodicity $\omega_j$, the time partition $\Pi_k^j$ of $\mathbf{X}^j$ can be further divided into a series of non-overlapping and non-empty subsets $\{\pi_{(k,s)}^j\}_{1 \leq s \leq \frac{\delta}{\omega_j}}$. For each $t \in [\tau_{\max} + 1, T]$, there exists $k \in [\omega_j]$ so that $t \in \Pi_k^j$ and further there exists $s \in [\frac{\delta}{\omega_j}]$ so that $t \in \pi_{(k,s)}^j$. $\pi_{(k,s)}^j$ can be written as:

$$\pi_{(k,s)}^j := \{t : \tau_{\max} + 1 \leq t \leq T, (t \bmod \omega_j) + 1 = k, (t \bmod \frac{\delta}{\omega_j}) + 1 = s\}. \tag{18}$$

With this definition, we have $\cup_{s=1}^{\frac{\delta}{\omega_j}}\pi_{(k,s)}^j = \Pi_k^j$. While time partition $\Pi_k^j$ guarantees that all variables in $\{X_t^j\}_{t\in\Pi_k^j}$ share the same causal mechanism, homogenous time partition $\pi_{(k,s)}^j$ guarantees that all variables in $\{X_t^j\}_{t>t',t\in\pi_{(k,s)}^j}$ share the same distribution where $t'$ represent the steps needed by the associated Markov chain to achieve equilibrium.

Fig.2 shows a partial causal graph for a 3-variate time series with Semi-Stationary SCM. $\tau_{\max} = 3$ means that the causal mechanisms start from $t = 4$, and the random variables with $t \in \{1, 2, 3\}$ are random noises. For the first time series $\mathbf{X}^1$, the periodicity $\omega_1$ is 3. And the periodicity of the time series $\mathbf{X}^2$ and $\mathbf{X}^3$ is 2 and 1, respectively. The periodicity of the whole time series $V$ is obtained by LCM$(3, 2, 1) = 6$. $\delta = \lceil\frac{\tau_{\max}+1}{\Omega}\rceil\Omega = \lceil\frac{3+1}{6}\rceil \times 6 = 1 \times 6 = 6$.

In Fig.2, periodicity $\omega_1 = 3$ means that the causal mechanisms repeat every three time points and hence there are three time partition subsets $\Pi_k^1, k \in [3]$. More specifically, $\Pi_1^1 = \{4, 7, 10, 13, ..., 4 + 3N, ...\}$, $\Pi_2^1 = \{5, 8, 11, 14, ..., 5 + 3N, ...\}$, $\Pi_3^1 = \{6, 9, 12, 15, ..., 6 + 3N, ...\}$ where $N \in \mathbb{N}^+$. Random variables $\{X_t^1\}$ with $t$ in the same time partition subset share the same causal mechanism. However, they may not share the same marginal distribution.

Still in Fig.2, based on the definition of homogenous time partition, time partition subset $\Pi_1^1$ for $\mathbf{X}^1$ can be further decomposed as $\pi_{(k=1,s=1)}^1 = \{4, 10, ..., 4 + \delta N, ...\}$, $\pi_{(k=1,s=2)}^1 = \{7, 13, ..., 7 + \delta N, ...\}$. where $s \in [\frac{\delta}{\omega_1}]$. After a long run $n$, $Z_n^1$ and $Z_{n+1}^1$ will eventually share the same distribution, that is, all the variables inside $Z_n^q$ will share the same joint or marginal distribution as the corresponding variables inside $Z_{n+1}^q$. To illustrate this, we assume that this Markov chain has already achieved its equilibrium at time point $t = 4$. Based on Eq.(12) and Eq.(17), we have:

$$p(\mathbf{X}_4, \mathbf{X}_5, ..., \mathbf{X}_9) = p(\mathbf{X}_{10}, \mathbf{X}_{11}, ..., \mathbf{X}_{15}) = p(\mathbf{X}_{16}, \mathbf{X}_{17}, ..., \mathbf{X}_{21}) = \cdots \tag{19}$$

From the identical joint distribution, we can further have:

$$p(X_4^1) = p(X_{10}^1) = p(X_{16}^1) = \cdots \tag{20}$$

as $X_4^1 \in \mathbf{X}_4$, $X_{10}^1 \in \mathbf{X}_{10}$ and $X_{16}^1 \in \mathbf{X}_{16}$.

Therefore, for sufficiently large values of $t$ ensuring that $Z_n^1$ has reached its stationary distribution, all variables within $\{X_t^j\}_{t\in\pi^j(k,s)}$ will share the same distribution.

In Fig.2, there are $6(=\delta)$ potentially different Markov chains $\{Z_n^q\}, q \in [\delta]$ in $V$. For any time window with length $\delta$, $\{\mathbf{X}_t, ..., \mathbf{X}_{t+\delta-1}\}$, there exists $q \in [\delta], n \in \mathbb{N}^+$, so that this time window can be completely included in $Z_n^q$. For instance, set $\{\mathbf{X}_5, \mathbf{X}_6, ..., \mathbf{X}_{10}\}$ is in $Z_1^2$, which is the first element of Markov chain $\{Z_n^2\}$.

Constructing Markov chains and applying the Irreducible and Aperiodic Markov Chain assumption (**A7**) enable us to obtain a consistent estimator for the conditional and joint distributions of interest.

## C.2 Consistent Estimator

The conditional distributions for variables in $\{X_t^j\}_{t\in\Pi_k^j}$ are the same, that is, $p(x_{t_1}^j|\text{Pa}(x_{t_1}^j)) = p(x_{t_2}^j|\text{Pa}(x_{t_2}^j))$, $\forall t_1, t_2 \in \Pi_k^j$. For simplicity, denote

$$p_{t\in\Pi_k^j}(x_t^j|\text{Pa}(x_t^j)) := p(x_t^j|\text{Pa}(x_t^j)), \ \forall t \in \Pi_k^j \tag{21}$$

Consider an indicator function such that $\mathbb{1}(x_t^j, \text{Pa}(x_t^j)) = 1$ if configuration $(x_t^j, \text{Pa}(x_t^j))$ has realized, otherwise $\mathbb{1}(x_t^j, \text{Pa}(x_t^j)) = 0$.

Since every $t \in \pi_{(k,s)}^j$ is apart from each other with $N\delta$ steps where $N \in \mathbb{N}^+$, and there must exist $q \in [\omega_j]$ and $n_1 \in \mathbb{N}^+$ so that $\{x_t^j, \text{Pa}(x_t^j)\} \in Z_{n_1}^q$, then for the same $q$, there must exist another $n_2$ so that $\{x_{t+N\delta}^j, \text{Pa}(x_{t+N\delta}^j)\} \in Z_{n_2}^q$. Hence, we have $\{\mathbb{1}(x_t^j, \text{Pa}(x_t^j))\}_{t\in\pi_{(k,s)}^j} = \{f(Z_{n_1(t)}^{q(t)})\}_{t\in\pi_{(k,s)}^j}$ with some function $f : \mathbb{R}^{n\times\delta} \to \mathbb{R}^1$ satisfying $E|f(Z_{n_1(t)}^{q(t)})| < \infty$. Since the value of $t$ determines $q$ and $n_1$, we use $q(t)$ and $n_1(t)$ to emphasize their relations. For large enough $t > t'$,

$\{\mathbb{1}(x_t^j, \mathrm{Pa}(x_t^j))\}_{t>t', t\in\pi_{(k,s)}^j}$ are identical samples where $t'$ is the time point needed by the associate Markov chain to achieve its equilibrium after $n_1(t')$ steps.

Without loss of generality, we assume $T$ is a multiple of $\delta$ all the time.

We can construct an estimator of $p(x_t^j, \mathrm{Pa}(x_t^j))$ with large enough $t$ as:

$$\hat{p}(x_t^j, \mathrm{Pa}(x_t^j)) = \frac{\delta}{T} \sum_{t\in\pi_{(k,s)}^j} \mathbb{1}(x_t^j, \mathrm{Pa}(x_t^j)) \tag{22}$$

where $k, s$ is determined by $t$ and there must exist one and only one $k, s$ satisfying $t \in \pi_{(k,s)}^j$. Now, we are going to show this estimator is consistent.

We first decompose the estimator into two parts: time point $t \le t'$ and $t > t'$, where $t'$ represents the time point when the equilibrium of the associated Markov chain is achieved.

$$\frac{\delta}{T} \sum_{t\in\pi_{(k,s)}^j} \mathbb{1}(x_t^j, \mathrm{Pa}(x_t^j)) \tag{23}$$

$$= \frac{\delta}{T}\left( \sum_{t\le t', t\in\pi_{(k,s)}^j} \mathbb{1}(x_t^j, \mathrm{Pa}(x_t^j)) + \sum_{t>t', t\in\pi_{(k,s)}^j} \mathbb{1}(x_t^j, \mathrm{Pa}(x_t^j)) \right) \tag{24}$$

$$= \frac{\delta}{T} \sum_{t\le t', t\in\pi_{(k,s)}^j} \mathbb{1}(x_t^j, \mathrm{Pa}(x_t^j)) + \frac{\delta}{T} \sum_{t>t', t\in\pi_{(k,s)}^j} \mathbb{1}(x_t^j, \mathrm{Pa}(x_t^j)) \tag{25}$$

$$= \frac{\delta}{T} \sum_{t\le t', t\in\pi_{(k,s)}^j} \mathbb{1}(x_t^j, \mathrm{Pa}(x_t^j)) + \frac{\delta}{T-t'}\frac{T-t'}{\delta}\frac{\delta}{T} \sum_{t>t', t\in\pi_{(k,s)}^j} \mathbb{1}(x_t^j, \mathrm{Pa}(x_t^j)) \tag{26}$$

$$= \frac{\delta}{T} \sum_{t\le t', t\in\pi_{(k,s)}^j} \mathbb{1}(x_t^j, \mathrm{Pa}(x_t^j)) + \frac{\delta}{T-t'}\frac{T-t'}{\delta}\frac{\delta}{T} \sum_{t>t', t\in\pi_{(k,s)}^j} \mathbb{1}(x_t^j, \mathrm{Pa}(x_t^j)) \tag{27}$$

$$= \frac{\delta}{T} \sum_{t\le t', t\in\pi_{(k,s)}^j} \mathbb{1}(x_t^j, \mathrm{Pa}(x_t^j)) + \frac{T-t'}{T}\left( \frac{\delta}{T-t'} \sum_{t>t', t\in\pi_{(k,s)}^j} \mathbb{1}(x_t^j, \mathrm{Pa}(x_t^j)) \right) \tag{28}$$

$$\tag{29}$$

Take a limit of Eq.(23), we have:

$$\lim_{T\to\infty} \frac{\delta}{T} \sum_{t\in\pi_{(k,s)}^j} \mathbb{1}(x_t^j, \mathrm{Pa}(x_t^j)) \tag{30}$$

$$= \lim_{T\to\infty} \frac{\delta}{T} \sum_{t\le t', t\in\pi_{(k,s)}^j} \mathbb{1}(x_t^j, \mathrm{Pa}(x_t^j)) + \lim_{T\to\infty} \frac{T-t'}{T}\left( \frac{\delta}{T-t'} \sum_{t>t', t\in\pi_{(k,s)}^j} \mathbb{1}(x_t^j, \mathrm{Pa}(x_t^j)) \right) \tag{31}$$

$$= 0 + \lim_{T\to\infty} \frac{T-t'}{T}\left( \frac{1}{n_1(T)-n_1(t')} \sum_{n_1(t)>n_1(t')}^{n_1(T)} f(Z_{n_1(t)}^{q(t)}) \right), \text{where } t > t', t \in \pi_{(k,s)}^j \tag{32}$$

$$\underset{\text{Birkhoff's Ergodic Theorem}}{=\!=\!=\!=\!=\!=\!=\!=\!=\!=\!=\!=} 0 + E\left( f(Z_{n_1(t)}^{q(t)}) \right) \tag{33}$$

$$= E\left( \mathbb{1}(x_t^j, \mathrm{Pa}(x_t^j)) \right), \text{where } t > t', t \in \pi_{(k,s)}^j \tag{34}$$

$$= p(x_t^j, \mathrm{Pa}(x_t^j)), \text{where } t > t', t \in \pi_{(k,s)}^j \tag{35}$$

Denote

$$p_{t\in\pi_{(k,s)}^j}(x_t^j, \mathrm{Pa}(x_t^j)) := p(x_t^j, \mathrm{Pa}(x_t^j)), \text{where } t > t', t \in \pi_{(k,s)}^j \tag{36}$$

Based on the definition of homogenous time partition and time partition, $p_{t\in\pi^j_{(k,s)}}(x^j_t|\mathrm{Pa}(x^j_t)) = p_{t\in\Pi^j_k}(x^j_t|\mathrm{Pa}(x^j_t))$, $\forall s \in [\frac{\delta}{\omega_j}]$.

Similar to Eq.(22), one estimator of $p_{t\in\Pi^j_k}(x^j_t|\mathrm{Pa}(x^j_t))$, $\forall k = [\omega_j]$ is

$$\hat{p}_{t\in\Pi^j_k}(x^j_t|\mathrm{Pa}(x^j_t)) = \frac{\sum_{t\in\Pi^j_k} \mathbb{1}(x^j_t, \mathrm{Pa}(x^j_t))}{\sum_{t\in\Pi^j_k} \mathbb{1}(\mathrm{Pa}(x^j_t))} \tag{37}$$

$$= \frac{\sum_{s=1}^{\frac{\delta}{\omega_j}} \sum_{t\in\pi^j_{(k,s)}} \mathbb{1}(x^j_t, \mathrm{Pa}(x^j_t))}{\sum_{s=1}^{\frac{\delta}{\omega_j}} \sum_{t\in\pi^j_{(k,s)}} \mathbb{1}(\mathrm{Pa}(x^j_t))} \tag{38}$$

$$= \frac{\sum_{s=1}^{\frac{\delta}{\omega_j}} \frac{T}{\delta} \sum_{t\in\pi^j_{(k,s)}} \mathbb{1}(x^j_t, \mathrm{Pa}(x^j_t))}{\sum_{s=1}^{\frac{\delta}{\omega_j}} \frac{T}{\delta} \sum_{t\in\pi^j_{(k,s)}} \mathbb{1}(\mathrm{Pa}(x^j_t))} \tag{39}$$

Take a limit of Eq.(37), we have:

$$\lim_{T\to\infty} \hat{p}_{t\in\Pi^j_k}(x^j_t|\mathrm{Pa}(x^j_t)) \tag{40}$$

$$\overset{\text{Eq.(35)}}{=\!=\!=\!=} \frac{\sum_{s=1}^{\frac{\delta}{\omega_j}} p_{t\in\pi^j_{(k,s)}}(x^j_t, \mathrm{Pa}(x^j_t))}{\sum_{s=1}^{\frac{\delta}{\omega_j}} p_{t\in\pi^j_{(k,s)}}(\mathrm{Pa}(x^j_t))} \tag{41}$$

$$= \frac{\sum_{s=1}^{\frac{\delta}{\omega_j}} p_{t\in\pi^j_{(k,s)}}(x^j_t|\mathrm{Pa}(x^j_t)) p_{t\in\pi^j_{(k,s)}}(\mathrm{Pa}(x^j_t))}{\sum_{s=1}^{\frac{\delta}{\omega_j}} p_{t\in\pi^j_{(k,s)}}(\mathrm{Pa}(x^j_t))} \tag{42}$$

$$\overset{p_{t\in\pi^j_{(k,s)}}(x^j_t|\mathrm{Pa}(x^j_t))\ \text{are same for all}\ s}{=\!=\!=\!=\!=\!=\!=\!=\!=\!=\!=} \frac{p_{t\in\Pi^j_k}(x^j_t|\mathrm{Pa}(x^j_t)) \sum_{s=1}^{\frac{\delta}{\omega_j}} p_{t\in\pi^j_{(k,s)}}(\mathrm{Pa}(x^j_t))}{\sum_{s=1}^{\frac{\delta}{\omega_j}} p_{t\in\pi^j_{(k,s)}}(\mathrm{Pa}(x^j_t))} \tag{43}$$

$$= p_{t\in\Pi^j_k}(x^j_t|\mathrm{Pa}(x^j_t)) \tag{44}$$

Hence, $\hat{p}_{t\in\Pi^j_k}(x^j_t|\mathrm{Pa}(x^j_t))$ is a consistent estimator of $p_{t\in\Pi^j_k}(x^j_t|\mathrm{Pa}(x^j_t))$.

Similarly, we construct an estimator of $p(x^j_t| \cup_h \mathrm{Pa}_h(x^j_t))$ where $t \in [T]$:

$$\hat{p}(x^j_t| \cup_h \mathrm{Pa}_h(x^j_t)) = \sum_t \mathbb{1}(x^j_t| \cup_h \mathrm{Pa}_h(x^j_t)) \tag{45}$$

$$= \frac{\sum_t \mathbb{1}(x^j_t, \cup_h\mathrm{Pa}_h(x^j_t))}{\sum_t \mathbb{1}(\cup_h\mathrm{Pa}_h(x^j_t))}. \tag{46}$$

We will prove that this estimator is converged as $T$ goes to infinity in Lemma D.2. Hence, it is a consistent estimator.

In this section, we have proved that $\hat{p}(x^j_t, \mathrm{Pa}(x^j_t))$ in Eq.(22) is a consistent estimator of $p(x^j_t, \mathrm{Pa}(x^j_t))$ using samples with $t$ in the same homogenous time partition subset and $\hat{p}(x^j_t|\mathrm{Pa}(x^j_t))$ in Eq.(37) is a consistent estimator of $p(x^j_t|\mathrm{Pa}(x^j_t))$ using samples with $t$ in the same time partition subset.

## D  Theorem

**Theorem D.1.** *Let $\widehat{\mathcal{G}}$ be the estimated graph using the Algorithm PCMCI$_\Omega$. Under assumptions A1-A7 and with an oracle (infinite sample size limit), we have that:*

$$\widehat{\mathcal{G}} = \mathcal{G} \tag{47}$$

*almost surely.*

**Lemma D.2.** *Denote that $\{\mathrm{Pa}_k(X_t^j)\}_{k\in[\omega_j]}$ contain the true and illusory parent sets, where $\omega_j$ is the true periodicity of $\mathbf{X}^j$. For any random variable $X_t^j$ with large enough $t$, under assumptions A1-A7 and with an oracle (infinite sample size limit), we have:*

$$p\left(p(X_t^j|\cup_{k=1}^{\omega_j}\mathrm{Pa}_k(X_t^j)) \neq p(X_t^j|\cup_{k=1}^{\omega_j}\mathrm{Pa}_k(X_t^j)\setminus y)\right) = 1,\ \forall y \in \cup_{k=1}^{\omega_j}\mathrm{Pa}_k(X_t^j) \tag{48}$$

*Here, $p(X_t^j|\cup_{k=1}^{\omega_j}\mathrm{Pa}_k(X_t^j)) = \lim_{T\to\infty}\hat{p}(X_t^j|\cup_{k=1}^{\omega_j}\mathrm{Pa}_k(X_t^j))$.*

*Proof.* We first prove that there exist a sequence of coefficients $\{\alpha_k\}_{k\in[\omega_j]}$ satisfying $\sum_{k=1}^{\omega_j}\alpha_k = 1$ so that:
$\forall$ configuration $\cup_h \mathrm{Pa}_h(x_t^j)$,

$$\hat{p}(x_t^j|\cup_h \mathrm{Pa}_h(x_t^j)) = \sum_{k=1}^{\omega_j}\alpha_k\hat{p}_k(x_t^j|\mathrm{Pa}(x_t^j)) \tag{49}$$

If this is correct, then $\hat{p}(x_t^j|\cup_h \mathrm{Pa}_h(x_t^j))$ would be a consistent estimator of $p(x_t^j|\cup_h \mathrm{Pa}_h(x_t^j))$. Based on Eq.(46), we have:

$$\hat{p}(x_t^j|\cup_h \mathrm{Pa}_h(x_t^j)) \tag{50}$$

$$= \frac{\sum_t \mathbb{1}(x_t^j,\cup_h\mathrm{Pa}_h(x_t^j))}{\sum_t \mathbb{1}(\cup_h\mathrm{Pa}_h(x_t^j))} \tag{51}$$

$$= \frac{\sum_t\sum_k \mathbb{1}(x_t^j,\cup_h\mathrm{Pa}_h(x_t^j))\mathbb{1}(t\in\Pi_k^j)}{\sum_t \mathbb{1}(\cup_h\mathrm{Pa}_h(x_t^j))} \tag{52}$$

$$= \sum_k\frac{\sum_t \mathbb{1}(x_t^j,\cup_h\mathrm{Pa}_h(x_t^j))\mathbb{1}(t\in\Pi_k^j)}{\sum_t \mathbb{1}(\cup_h\mathrm{Pa}_h(x_t^j))} \tag{53}$$

$$= \sum_k\frac{\sum_{t\in\Pi_k^j} \mathbb{1}(x_t^j,\cup_h\mathrm{Pa}_h(x_t^j))}{\sum_t \mathbb{1}(\cup_h\mathrm{Pa}_h(x_t^j))} \tag{54}$$

$$= \sum_k\left(\frac{\sum_{t\in\Pi_k^j} \mathbb{1}(x_t^j,\cup_h\mathrm{Pa}_h(x_t^j))}{\sum_t \mathbb{1}(\cup_h\mathrm{Pa}_h(x_t^j))}\frac{\sum_t \mathbb{1}(\cup_h\mathrm{Pa}_h(x_t^j))}{\sum_{t\in\Pi_k^j} \mathbb{1}(\cup_h\mathrm{Pa}_h(x_t^j))}\frac{\sum_{t\in\Pi_k^j} \mathbb{1}(\cup_h\mathrm{Pa}_h(x_t^j))}{\sum_t \mathbb{1}(\cup_h\mathrm{Pa}_h(x_t^j))}\right) \tag{55}$$

$$= \sum_k\left(\frac{\sum_{t\in\Pi_k^j} \mathbb{1}(x_t^j,\cup_h\mathrm{Pa}_h(x_t^j))}{\sum_{t\in\Pi_k^j} \mathbb{1}(\cup_h\mathrm{Pa}_h(x_t^j))}\frac{\sum_{t\in\Pi_k^j} \mathbb{1}(\cup_k\mathrm{Pa}_{t\in\Pi_k^j}(x_t^j))}{\sum_t \mathbb{1}(\cup_h\mathrm{Pa}_h(x_t^j))}\right) \tag{56}$$

$$= \sum_k\left(\hat{p}_{t\in\Pi_k^j}(x_t^j|\cup_h \mathrm{Pa}_h(x_t^j))\frac{\sum_{t\in\Pi_k^j} \mathbb{1}(\cup_h\mathrm{Pa}_h(x_t^j))}{\sum_t \mathbb{1}(\cup_h\mathrm{Pa}_h(x_t^j))}\right) \tag{57}$$

$$= \sum_k\left(\hat{p}_{t\in\Pi_k^j}(x_t^j|\mathrm{Pa}(x_t^j),\cup_h\mathrm{Pa}_h(x_t^j)\setminus\mathrm{Pa}(x_t^j))\frac{\sum_{t\in\Pi_k^j} \mathbb{1}(\cup_h\mathrm{Pa}_h(x_t^j))}{\sum_t \mathbb{1}(\cup_h\mathrm{Pa}_h(x_t^j))}\right) \tag{58}$$

$$= \sum_k\left(\hat{p}_{t\in\Pi_k^j}(x_t^j|\mathrm{Pa}(x_t^j))\frac{\sum_{t\in\Pi_k^j} \mathbb{1}(\cup_h\mathrm{Pa}_h(x_t^j))}{\sum_t \mathbb{1}(\cup_h\mathrm{Pa}_h(x_t^j))}\right) \tag{59}$$

$$= \sum_k\alpha_k(T)\hat{p}_{t\in\Pi_k^j}(x_t^j|\mathrm{Pa}(x_t^j)), \tag{60}$$

$$\text{where } \alpha_k(T) = \frac{\sum_{t\in\Pi_k^j} \mathbb{1}(\cup_h\mathrm{Pa}_h(x_t^j))}{\sum_t \mathbb{1}(\cup_h\mathrm{Pa}_h(x_t^j))}. \tag{61}$$

Using the same logic in Eq.(30)-(35), we can decompose the numerator and denominator of $\alpha_k$ with homogenous time partition until each component converges to a stationary distribution.

$$\alpha_k(T) = \frac{\sum_{t \in \Pi_k^j} \mathbb{1}(\cup_h \mathrm{Pa}_h(x_t^j))}{\sum_k \sum_{t \in \Pi_k^j} \mathbb{1}(\cup_h \mathrm{Pa}_h(x_t^j))} \tag{62}$$

$$= \frac{\sum_{s=1}^{\frac{\delta}{\omega_j}} \frac{T}{\delta} \sum_{t \in \pi_{(k,s)}^j} \mathbb{1}(\cup_h \mathrm{Pa}_h(x_t^j))}{\sum_k \sum_{s=1}^{\frac{\delta}{\omega_j}} \frac{T}{\delta} \sum_{t \in \pi_{(k,s)}^j} \mathbb{1}(\cup_h \mathrm{Pa}_h(x_t^j))} \tag{63}$$

$$\lim_{T \to \infty} \alpha_k(T) = \frac{\sum_{s=1}^{\frac{\delta}{\omega_j}} p_{t \in \pi_{(k,s)}^j}(\cup_h \mathrm{Pa}_h(x_t^j))}{\sum_k \sum_{s=1}^{\frac{\delta}{\omega_j}} p_{t \in \pi_{(k,s)}^j}(\cup_h \mathrm{Pa}_h(x_t^j))} \tag{64}$$

Without loss of generality, assume $y \in \mathrm{Pa}(x_t^j)$, where $t \in \Pi_j^1$ and $y \notin \mathrm{Pa}(x_t^j)$, where $t \notin \Pi_j^1$ . Then we have

$$\hat{p}(x_t^j | \cup_h \mathrm{Pa}_h(x_t^j) \setminus y) = \frac{\sum_t \mathbb{1}(x_t^j, \cup_h \mathrm{Pa}_h(x_t^j) \setminus y)}{\sum_t \mathbb{1}(\cup_h \mathrm{Pa}_h(x_t^j) \setminus y)} \tag{65}$$

$$= \sum_{k=2}^{\omega_j} \left( \hat{p}_{t \in \Pi_k^j}(x_t^j | \cup_h \mathrm{Pa}_h(x_t^j) \setminus y) \frac{\sum_{t \in \Pi_k^j} \mathbb{1}(\cup_h \mathrm{Pa}_h(x_t^j) \setminus y)}{\sum_t \mathbb{1}(\cup_h \mathrm{Pa}_h(x_t^j) \setminus y)} \right) \tag{66}$$

$$+ \frac{\sum_{t \in \Pi_1^j} \mathbb{1}(x_t^j, \cup_h \mathrm{Pa}_h(x_t^j) \setminus y)}{\sum_{t \in \Pi_1^j} \mathbb{1}(\cup_h \mathrm{Pa}_h(x_t^j) \setminus y)} \frac{\sum_{t \in \Pi_1^j} \mathbb{1}(\cup_h \mathrm{Pa}_h(x_t^j) \setminus y)}{\sum_t \mathbb{1}(\cup_h \mathrm{Pa}_h(x_t^j) \setminus y)}$$

$$= \sum_{k=2}^{\omega_j} \beta_k(T) \hat{p}_{t \in \Pi_k^j}(x_t^j | \mathrm{Pa}(x_t^j)) + \beta_1(T) \hat{p}_{t \in \Pi_1^j}(x_t^j | \mathrm{Pa}(x_t^j) \setminus y) \tag{67}$$

$$\text{where } \beta_k(T) = \frac{\sum_{t \in \Pi_k^j} \mathbb{1}(\cup_h \mathrm{Pa}_h(x_t^j) \setminus y)}{\sum_t \mathbb{1}(\cup_h \mathrm{Pa}_h(x_t^j) \setminus y)} \tag{68}$$

Similarly, we have:

$$\beta_k(T) = \frac{\sum_{t \in \Pi_k^j} \mathbb{1}(\cup_h \mathrm{Pa}_h(x_t^j) \setminus y)}{\sum_k \sum_{t \in \Pi_k^j} \mathbb{1}(\cup_h \mathrm{Pa}_h(x_t^j) \setminus y)} \tag{69}$$

$$= \frac{\sum_{s=1}^{\frac{\delta}{\omega_j}} \frac{T}{\delta} \sum_{t \in \pi_{(k,s)}^j} \mathbb{1}(\cup_h \mathrm{Pa}_h(x_t^j) \setminus y)}{\sum_k \sum_{s=1}^{\frac{\delta}{\omega_j}} \frac{T}{\delta} \sum_{t \in \pi_{(k,s)}^j} \mathbb{1}(\cup_h \mathrm{Pa}_h(x_t^j) \setminus y)} \tag{70}$$

$$\lim_{T \to \infty} \beta_k(T) = \frac{\sum_{s=1}^{\frac{\delta}{\omega_j}} p_{t \in \pi_{(k,s)}^j}(\cup_h \mathrm{Pa}_h(x_t^j) \setminus y)}{\sum_k \sum_{s=1}^{\frac{\delta}{\omega_j}} p_{t \in \pi_{(k,s)}^j}(\cup_h \mathrm{Pa}_h(x_t^j) \setminus y)} \tag{71}$$

Proving $p(x_t^j | \cup_h \mathrm{Pa}_h(x_t^j)) \neq p(x_t^j | \cup_h \mathrm{Pa}_h(x_t^j) \setminus y)$ is equal to proving:

$$p(x_t^j | \cup_h \mathrm{Pa}_h(x_t^j)) - p(x_t^j | \cup_h \mathrm{Pa}_h(x_t^j) \setminus y) \neq 0 \tag{72}$$

Substitutes Eq.(60) and Eq.(67) in Eq.(72), we have the following derivation:

$$p(x_t^j \mid \cup_h \mathrm{Pa}_h(x_t^j)) - p(x_t^j \mid \cup_h \mathrm{Pa}_h(x_t^j) \setminus y) \tag{73}$$

$$= \lim_{T \to \infty} \left( \sum_{k=1}^{\omega_j} \alpha_k(T) \hat{p}_{t \in \Pi_k^j}(x_t^j \mid \mathrm{Pa}(x_t^j)) \right) - \tag{74}$$

$$\lim_{T \to \infty} \left( \sum_{k=2}^{\omega_j} \beta_k(T) \hat{p}_{t \in \Pi_k^j}(x_t^j \mid \mathrm{Pa}(x_t^j)) + \beta_1(T) \hat{p}_{t \in \Pi_1^j}(x_t^j \mid \mathrm{Pa}(x_t^j) \setminus y) \right)$$

$$= \sum_{k=1}^{\omega_j} \frac{\sum_{s=1}^{\frac{\delta}{\omega_j}} p_{t \in \pi_{(k,s)}^j}(\cup_h \mathrm{Pa}_h(x_t^j))}{\sum_k \sum_{s=1}^{\frac{\delta}{\omega_j}} p_{t \in \pi_{(k,s)}^j}(\cup_h \mathrm{Pa}_h(x_t^j))} p_{t \in \Pi_k^j}(x_t^j \mid \mathrm{Pa}(x_t^j)) \tag{75}$$

$$- \left( \sum_{k=2}^{\omega_j} \frac{\sum_{s=1}^{\frac{\delta}{\omega_j}} p_{t \in \pi_{(k,s)}^j}(\cup_h \mathrm{Pa}_h(x_t^j) \setminus y)}{\sum_k \sum_{s=1}^{\frac{\delta}{\omega_j}} p_{t \in \pi_{(k,s)}^j}(\cup_h \mathrm{Pa}_h(x_t^j) \setminus y)} p_{t \in \Pi_k^j}(x_t^j \mid \mathrm{Pa}(x_t^j)) \right.$$

$$\left. + \frac{\sum_{s=1}^{\frac{\delta}{\omega_j}} p_{t \in \Pi_{(1,s)}^j}(\cup_h \mathrm{Pa}_h(x_t^j) \setminus y)}{\sum_k \sum_{s=1}^{\frac{\delta}{\omega_j}} p_{t \in \pi_{(k,s)}^j}(\cup_h \mathrm{Pa}_h(x_t^j) \setminus y)} p_{t \in \Pi_1^j}(x_t^j \mid \mathrm{Pa}(x_t^j) \setminus y) \right)$$

After equating the denominators, the numerator is:

$$\left( \sum_{k=1}^{\omega_j} \sum_{s=1}^{\frac{\delta}{\omega_j}} p_{t \in \pi_{(k,s)}^j}(\cup_h \mathrm{Pa}_h(x_t^j)) p_{t \in \Pi_k^j}(x_t^j \mid \mathrm{Pa}(x_t^j)) \right) \left( \sum_{k=1}^{\omega_j} \sum_{s=1}^{\frac{\delta}{\omega_j}} p_{t \in \pi_{(k,s)}^j}(\cup_h \mathrm{Pa}_h(x_t^j) \setminus y) \right)$$

$$- \left( \sum_{k=2}^{\omega_j} \sum_{s=1}^{\frac{\delta}{\omega_j}} p_{t \in \pi_{(k,s)}^j}(\cup_h \mathrm{Pa}_h(x_t^j) \setminus y) p_{t \in \Pi_k^j}(x_t^j \mid \mathrm{Pa}(x_t^j)) \right.$$

$$\left. + \sum_{s=1}^{\frac{\delta}{\omega_j}} p_{t \in \Pi_{(1,s)}^j}(\cup_h \mathrm{Pa}_h(x_t^j) \setminus y) p_{t \in \Pi_1^j}(x_t^j \mid \mathrm{Pa}(x_t^j) \setminus y) \right)$$

$$\times \left( \sum_{k=1}^{\omega_j} \sum_{s=1}^{\frac{\delta}{\omega_j}} p_{t \in \pi_{(k,s)}^j}(\cup_h \mathrm{Pa}_h(x_t^j)) \right) \tag{76}$$

For the sake of simplicity, denote

$$a_k := \sum_{s=1}^{\frac{\delta}{\omega_j}} p_{t \in \pi_{(k,s)}^j}(\cup_h \mathrm{Pa}_h(x_t^j)) \tag{77}$$

$$b_k := \sum_{s=1}^{\frac{\delta}{\omega_j}} p_{t \in \pi_{(k,s)}^j}(\cup_h \mathrm{Pa}_h(x_t^j) \setminus y) \tag{78}$$

$$c_k := p_{t \in \Pi_k^j}(x_t^j \mid \mathrm{Pa}(x_t^j)) \tag{79}$$

$$c_1' := p_{t \in \Pi_1^j}(x_t^j \mid \mathrm{Pa}(x_t^j) \setminus y) \tag{80}$$

After substituting the simple notations in Eq.(76):

$$\left(\sum_{k=1}^{\omega_j} a_k c_k\right)\left(\sum_{k=1}^{\omega_j} b_k\right) - \left(\sum_{k=2}^{\omega_j} b_k c_k + b_1 c_1'\right)\left(\sum_{k=1}^{\omega_j} a_k\right) \tag{81}$$

$$= \sum_{k=1}^{\omega_j}(c_k - c_{1'})a_k b_1 + \sum_{k=1}^{\omega_j}\sum_{i>1, i\neq k}^{\omega_j}(c_k - c_i)a_k b_i \tag{82}$$

$$= b_1\sum_{k=1}^{\omega_j} c_k a_k - c_{1'} b_1 \sum_{k=1}^{\omega_j} a_k + \sum_{k=1}^{\omega_j} c_k a_k \sum_{i>1,i\neq k}^{\omega_j} b_i - \sum_{k=1}^{\omega_j} a_k \sum_{i>1,i\neq k}^{\omega_j} c_i b_i \tag{83}$$

Define

$$V_t = \{\mathbf{X}_{t'} | 0 < t' < t\} \tag{84}$$

That is, $V_t$ contains all the nodes before time point $t$.

Denote $\{b_{t_i}\}_{i\in[n]} = \cup_h \mathrm{Pa}_h(x_t^j)$ and assume $\{b_{t_i}\}_{1\leq i\leq n_1 < n} = \mathrm{Pa}_1(x_t^j)$, where $t \in \Pi_{(k,s)}^j$

We express $p_{t\in\Pi_{(k,s)}^j}(\cup_h \mathrm{Pa}_h(x_t^j))$ by marginalizing all other random variables occurring before the latest variables in $\cup_h \mathrm{Pa}_h(x_t^j)$ and utilizing the Causal Markov assumption (**A2**):

$$p_{t\in\Pi_{(k,s)}^j}(\cup_h \mathrm{Pa}_h(x_t^j)) \tag{85}$$

$$= p(\cup_h \mathrm{Pa}_h(x_t^j)|t \in \Pi_{(k,s)}^j) \tag{86}$$

$$= \sum_{V_h\setminus\{b_{t_i}\}_{i\in[n]}} p(b_{t_1}, b_{t_2}, ... b_{t_n}, V_h \setminus \{b_{t_i}\}_{i\in[n]}|h = \max\{t_i, 1 \leq i \leq n\}, t \in \Pi_{(k,s)}^j) \tag{87}$$

$$= \sum_{\{\mathrm{Pa}(b_{t_i})\}_{i\in[n]}} p(b_{t_i}|\mathrm{Pa}(b_{t_i})) \sum_{V_{\tau_{\max}}} \sum_{\tau_{\max}<t'\leq h} \sum_{j\in[n]} \sum_{x_{t'}^j, \mathrm{Pa}(x_{t'}^j)} p(x_{t'}^j|\mathrm{Pa}(x_{t'}^j))p(V_{\tau_{\max}}) \tag{88}$$

Note that $x_{t'}^j \in V_h \setminus \{b_{t_i}\}_{i\in[n]}$.

This joint distribution is now represented by conditional distributions of one related variable given its parents.

Similarly, assume $y = b_{t_1}$, we have

$$p_{t\in\Pi_{(k,s)}^j}(\cup_h \mathrm{Pa}_h(x_t^j) \setminus y) \tag{89}$$

$$= p_{t\in\Pi_{(k,s)}^j}(\cup_h \mathrm{Pa}_h(x_t^j) \setminus b_{t_1}) \tag{90}$$

$$= p(\cup_h \mathrm{Pa}_h(x_t^j \setminus y)|t \in \Pi_{(k,s)}^j) \tag{91}$$

$$= \sum_{V_{t_n}\setminus\{b_{t_i}\}_{i\neq 1}} p(b_{t_2}, ... b_{t_{n-1}}, b_{t_n}, V_{t_n} \setminus \{b_{t_i}\}_{i\neq 1}|h = \max\{t_i, 2 \leq i \leq n\}) \tag{92}$$

$$= \sum_{\{\mathrm{Pa}(b_{t_i})\}_{i\neq 1}} p(b_{t_i}|\mathrm{Pa}(b_{t_i})) \sum_{V_{\tau_{\max}}} \sum_{\tau_{\max}<t'\leq h} \sum_{j\in[n]} \sum_{x_{t'}^j, \mathrm{Pa}(x_{t'}^j)} p(x_{t'}^j|\mathrm{Pa}(x_{t'}^j))p(V_{\tau_{\max}}) \tag{93}$$

Note that $x_{t'}^j \in V_{t_n} \setminus \{b_{t_i}\}_{i\neq 1}$.

$p_{t\in\Pi_1^j}(x_t^j|\mathrm{Pa}(x_t^j)\setminus y)$ can also be represented by those conditional distributions based on Bayes rule.

$$p_{t\in\Pi_1^j}(x_t^j|\mathrm{Pa}(x_t^j)\setminus b_{t_1}) \tag{94}$$

$$= \frac{p_{t\in\Pi_1^j}(x_t^j, b_{t_2},..,b_{t_{n_1}})}{p_{t\in\Pi_1^j}(b_{t_2},...,b_{t_{n_1}})} \tag{95}$$

$$= \frac{\sum_{b_{t_1}} p_{t\in\Pi_1^j}(x_t^j, b_{t_1},...,b_{t_{n_1}})}{\sum_{b_{t_1}} p_{t\in\Pi_1^j}(b_{t_1},...,b_{t_{n_1}})} \tag{96}$$

$$= \frac{\sum_{b_{t_1}} p_{t\in\Pi_1^j}(x_t^j, \mathrm{Pa}_1(x_t^j))}{\sum_{b_{t_1}} p_{t\in\Pi_1^j}(\mathrm{Pa}_1(x_t^j))} \tag{97}$$

$$= \frac{\sum_{b_{t_1}} p(x_t^j|\mathrm{Pa}_1(x_t^j))p_{t\in\Pi_1^j}(b_{t_1},...,b_{t_{n_1}})}{\sum_{b_{t_1}} p_{t\in\Pi_1^j}(b_{t_1},...,b_{t_{n_1}})} \tag{98}$$

$$= \frac{\sum_{b_{t_1}} p(x_t^j|\mathrm{Pa}_1(x_t^j)) \sum_{V_h\setminus\{b_{t_{i\in[n_1]}}\}} p_{t\in\Pi_1^j}(b_{t_1},...b_{t_{n_1}}, V_h\setminus\{b_{t_{i\in[n_1]}}\}|h=\max\{t_{i\in[n_1]}\})}{\sum_{b_{t_1}} \sum_{V_h\setminus\{b_{t_{i\in[n_1]}}\}} p_{t\in\Pi_1^j}(b_{t_1},...b_{t_{n_1}}, V_h\setminus\{b_{t_{i\in[n_1]}}\}|h=\max\{t_{i\in[n_1]}\})} \tag{99}$$

$$= \frac{\sum_{b_{t_1}} AB}{\sum_{b_{t_1}} CD} \tag{100}$$

where

$$A = p(x_t^j|\mathrm{Pa}_1(x_t^j)) \sum_{\{\mathrm{Pa}(b_{t_i})\}_{i\in[n_1]}} p(b_{t_i}|\mathrm{Pa}(b_{t_i})) \tag{101}$$

$$B = \sum_{V_{\tau_{\max}}} \sum_{\tau_{\max}<t'\le h} \sum_{j\in[n]} \sum_{x_{t'}^j, \mathrm{Pa}(x_{t'}^j)} p(x_{t'}^j|\mathrm{Pa}(x_{t'}^j))p(V_{\tau_{\max}}) \tag{102}$$

$$C = \sum_{\{\mathrm{Pa}(b_{t_i})\}_{i\in[n_1]}} p(b_{t_i}|\mathrm{Pa}(b_{t_i})) \tag{103}$$

$$D = \sum_{V_{\tau_{\max}}} \sum_{\tau_{\max}<t'\le h} \sum_{j\in[n]} \sum_{x_{t'}^j, \mathrm{Pa}(x_{t'}^j)} p(x_{t'}^j|\mathrm{Pa}(x_{t'}^j))p(V_{\tau_{\max}}) \tag{104}$$

Note that $t\in\Pi_1^j$ for distributions in above section from Eq.(94) to Eq.(104) and that $x_{t'}^j \in V_h\setminus\{b_{t_i}\}_{i\in[n_1]}$.

Hence, every term in Eq.(83) can be expressed as a function of those conditional distributions. Substituting Eq.(88), Eq.(93) and Eq.(100) in Eq.(83), we have a polynomial equation only composed of conditional distributions $\{p(x_{t'}^j|\mathrm{Pa}(x_{t'}^j))\}_{j\in[n],t'\le t}$ except the joint distribution of the starting points $p(V_{\tau_{\max}})$. Note that the conditional distributions of variables in $\{X_t^j\}_{t\in\Pi_k^j}, j\in[n], k\in[\omega_j]$ are the same. Since sets do not allow duplicate values, set $\{p(x_{t'}^j|\mathrm{Pa}(x_{t'}^j))\}_{j\in[n],t'\le t}$ contains only different conditional distributions. There should be potentially total $\sum_{j=1}^n \omega_j$ different causal mechanisms. The total number of conditional probabilities should be jointly determined by the number of causal mechanisms and also the number of realizations that variables can take. After adjusting those conditional distributions by the linear restriction $\sum_y p(x|y)=1$, all components in the set $\{p(x_{t'}^j|\mathrm{Pa}(x_{t'}^j))\}_{j\in[n],t'\le t}$ are mutually independent, and $p(V_{\tau_{\max}})$ is also independent of all the causal mechanisms because the first starting points are random noises. That is, upon adjustments, all the terms in Eq.(83) should be rendered independent of each other, without any imposed constraints across them.

After expanding all the summations in Eq.(83), the coefficients of this polynomial equation are either 1 or $-1$. Each coefficient is accompanied by one unique monomial as index $(k,s)$ in the joint distribution $p_{t\in\pi_{k,s}^j}$ determined a unique product of conditional distributions, i.e., with a different pair of $(k,s)$, the product should be different. Considering all random and independent conditional

distributions in $\{p\big(x_{t'}^j|\mathrm{Pa}(x_{t'}^j)\big)\}_{j\in[n],t'\leq t}$, the polynomial is not identically zero, and the probability of choosing a root of this polynomial is zero.

Denote the polynomial equation in Eq.(83) as A, we have:

$$p(A=0)=0 \tag{105}$$

Back to the original Eq.(73), we finally have $p\Big(p(x_t^j|\cup_h \mathrm{Pa}_h(x_t^j)) \neq p(x_t^j|\cup_h \mathrm{Pa}_h(x_t^j) \setminus y)\Big) = 1,\ \forall y \in \cup_h\mathrm{Pa}_h(x_t^j)$.

$\square$

**Lemma D.3.** *Let $\widehat{S\mathrm{Pa}}(\mathbf{X}_t^j)$ denote the estimated superset of parent set for $\mathbf{X}^j \in V$ obtained from the Algorithm B1 (line 2). $\{\mathrm{Pa}_k(X_t^j)\}_{k\in[\omega_j]}$ contain the true and illusory parent sets, where $\omega_j$ is the true periodicity of $\mathbf{X}^j$. Under assumptions A1-A7 and with an oracle (infinite sample size limit), we have:*

$$\cup_{k=1}^{\omega_j}\mathrm{Pa}_k(X_t^j) \subseteq \widehat{S\mathrm{Pa}}(X_t^j),\ \forall t \in [\tau_{\max}+1, T]$$

*almost surely.*

*Proof.* Assume the contrary, i.e., there exists $s \in \cup_k\mathrm{Pa}_k(X_t^j) \setminus \widehat{S\mathrm{Pa}}(X_t^j)$. From Lemma D.2, we have $X_t^j \not\perp\!\!\!\perp s \Big| \cup_{k=1}^{\omega_j}\mathrm{Pa}_k(X_t^j) \setminus s$. By the Definition 2.4, we have $\mathrm{Pa}(X_t^j) \subset \cup_{k=1}^{\omega_j}\mathrm{Pa}_k(X_t^j)$. If $s \notin \mathrm{Pa}(X_t^j)$, by the causal Markov property (**A2**), the dependence relation can not be true, because $s$ is a non-descendant of $X_t^j$. If $s \in \mathrm{Pa}(X_t^j)$, our Algorithm would have concluded that $X_t^j \not\perp\!\!\!\perp s \Big| \widehat{S\mathrm{Pa}}(X_t^j)$ (line 2) with a consistent CI test, evident from the causal Markov property, contradicting our assumption. Hence, the lemma. $\square$

**Lemma D.4.** *Let $\widehat{Pa}(X_t^j)$ denote the estimated parent set for $\mathbf{X}^j \in V$ obtained from the Algorithm B1 (line 19) assuming that true $\omega_j$ has obtained (line 17). $\{\mathrm{Pa}_k(X_t^j)\}_{k\in[\omega_j]}$ contain the true and illusory parent sets. Under assumptions A1-A7 and with an oracle (infinite sample size limit), we have:*

$$\widehat{Pa}(X_t^j) = Pa(X_t^j),\ \forall t \in [\tau_{\max}+1, T] \tag{106}$$

*almost surely.*

*Proof.* From Lemma D.3,

$$Pa(X_t^j) \subset \cup_{k=1}^{\omega_j}\mathrm{Pa}_k(X_t^j) \subseteq \widehat{S\mathrm{Pa}}(X_t^j),\ \forall t \in [\tau_{\max}+1, T], j \in [n] \tag{107}$$

In Runge et al. [2019], the author proved $\widehat{Pa}(X_t^j) = Pa(X_t^j)$ if we run PCMCI on stationary time series. Using the same logic, we have the following proof.

Suppose $X_{t-\tau}^i \notin \widehat{Pa}(X_t^j)$ but $X_{t-\tau}^i \in Pa(X_t^j)$. With a consistent conditional independence test and correct time partition, the $MCI$ test (line 8 in Algorithm B1) will remove $X_{t-\tau}^i$ from $\widehat{Pa}_{\omega_j}(X_t^j)$ if and only if:

$$X_{t-\tau}^i \perp\!\!\!\perp X_t^j \Big| \widehat{SPa}(X_t^j) \setminus \{X_{t-\tau}^i\}, \widehat{SPa}(X_{t-\tau}^i) \tag{108}$$

Based on Eq.(107), the rule is equivalent to removing $X_{t-\tau}^i$ from $\widehat{Pa}_{\omega_j}(X_t^j)$ if and only if:

$$X_{t-\tau}^i \perp\!\!\!\perp X_t^j \Bigg| \Bigg\{ Pa(X_t^j) \setminus \{X_{t-\tau}^i\}, Pa(X_{t-\tau}^i),$$

$$\widehat{SPa}(X_t^j) \setminus (Pa(X_t^j) \cup \{X_{t-\tau}^i\}), \widehat{SPa}(X_{t-\tau}^i) \setminus Pa(X_{t-\tau}^i) \Bigg\} \tag{109}$$

$$\Rightarrow X_{t-\tau}^i \perp\!\!\!\perp X_t^j \Big| Pa(X_t^j) \setminus \{X_{t-\tau}^i\}, Pa(X_{t-\tau}^i) \tag{110}$$

Based on Causal Markov Condition assumption (**A2**) and Faithfulness Condition (**A3**), from Eq.(110) we have $X_{t-\tau}^i \notin Pa(X_t^j)$. In other words, if $X_{t-\tau}^i \notin \widehat{Pa}(X_t^j)$ then $X_{t-\tau}^i \notin Pa(X_t^j)$. That is, $Pa(X_t^j) \subseteq \widehat{Pa}(X_t^j)$

Suppose $X_{t-\tau}^i \in \widehat{Pa}(X_t^j)$ but $X_{t-\tau}^i \notin Pa(X_t^j)$. By the contraposition of Faithfulness (**A1**), we know that $X_{t-\tau}^i \not\perp\!\!\!\perp X_t^j \big| \widehat{Pa}(X_t^j) \setminus \{X_{t-\tau}^i\}, \widehat{Pa}(X_{t-\tau}^i)$. Denote $W = \{\widehat{Pa}(X_t^j) \setminus \{Pa(X_t^j), X_{t-\tau}^i\}\} \cup \{\widehat{Pa}(X_{t-\tau}^i) \setminus Pa(X_{t-\tau}^i)\}$. Since $X_{t-\tau}^i \notin Pa(X_t^j)$, based on Causal Markov Condition assumption (**A2**),

$$W \cup X_{t-\tau}^i \perp\!\!\!\perp X_t^j \big| Pa(X_t^j)$$

$$\Rightarrow W \cup X_{t-\tau}^i \perp\!\!\!\perp X_t^j \big| Pa(X_t^j), Pa(X_{t-\tau}^i)$$

$$\xrightarrow{\text{Weak Union}} X_{t-\tau}^i \perp\!\!\!\perp X_t^j \big| \{Pa(X_t^j), Pa(X_{t-\tau}^i)\} \cup W$$

$$\Rightarrow X_{t-\tau}^i \perp\!\!\!\perp X_t^j \big| \widehat{Pa}(X_t^j) \setminus \{X_{t-\tau}^i\}, \widehat{Pa}(X_{t-\tau}^i)$$

This is contrary to the assumption so that there is no such $X_{t-\tau}^i$ satisfying $X_{t-\tau}^i \in \widehat{Pa}(X_t^j)$ but $X_{t-\tau}^i \notin Pa(X_t^j)$. In other words, if $X_{t-\tau}^i \in \widehat{Pa}(X_t^j)$, then $X_{t-\tau}^i \in Pa(X_t^j)$. That is, $\widehat{Pa}(X_t^j) \subseteq Pa(X_t^j)$. Combined with the previous conclusion that $Pa(X_t^j) \subseteq \widehat{Pa}(X_t^j)$, we have $\widehat{Pa}(X_t^j) = Pa(X_t^j)$.

$\square$

Based on Lemma D.2, Lemma D.3 and Lemma D.4, we can identify the true $\omega_j$ for $\mathbf{X}^j$ through Lemma D.5.

**Lemma D.5.** *Let $\omega_j$ denote the true periodicity for $\mathbf{X}^j \in V$ and $\widehat{\mathrm{Pa}}(X_{t\in\Pi_k^j}^j)$ denote the estimated parent set for $X_t^j$ obtained from Algorithm B1 where $t \in \Pi_k^j$. Define:*

$$\widehat{\omega}_j = \arg \min_{\omega \in [\omega_{ub}]} \max_{k \in [\omega]} |\widehat{\mathrm{Pa}}(X_{t\in\Pi_k^j}^j)| \tag{111}$$

*Under assumptions A1-A7 and with an oracle (infinite sample limit), we have that $\hat{\omega}_j = \omega_j$, $\forall j \in [n]$ almost surely.*

*Proof.* Assume the contrary that $\hat{\omega}_j \neq \omega_j$, then in the Algorithm B1, we have an incorrect time partition $\widehat{\Pi}^j$. Hence, CI tests that are performed use samples with different causal mechanisms. $\hat{p}(X_t^j | \cup_{k=1}^{\omega_j} \mathrm{Pa}_k(X_t^j))$ in Eq.(50) is estimated from a mixture of two or more time partition subsets, say $\Pi_1^j$ and $\Pi_2^j$. We can apply Lemma D.2 where $\cup_{k=1}^{\omega_j} \mathrm{Pa}_k(X_t^j)$ is replaced by $\cup_{k=1}^2 \mathrm{Pa}_k(X_t^j)$ and then in Lemma D.3, $\widehat{S\mathrm{Pa}}(X_t^j)$ is replaced by $\widehat{\mathrm{Pa}}_{\hat{\omega}_j}(X_t^j)$ and hence $\cup_{k=1}^2 \mathrm{Pa}_k(X_t^j) \subseteq \widehat{\mathrm{Pa}}_{\hat{\omega}_j}(X_t^j)$ where $\widehat{\mathrm{Pa}}_{\hat{\omega}_j}(X_t^j)$ is obtained from samples with $t$ from the mixture of two different partition subsets (line 8). Hence, with $\hat{\omega}_j$, $|\widehat{\mathrm{Pa}}_{\hat{\omega}_j}(X_t^j)| \geq |\cup_{k=1}^2 \mathrm{Pa}_k(X_t^j)|$ using mixture samples $t \in \cup_{k=1}^2 \Pi_k^j$. However, with true $\omega_j$, we have $|\widehat{\mathrm{Pa}}_{\omega_j}(X_t^j)| = |\mathrm{Pa}(X_t^j)|$ based on Lemma D.4. With Assumption **A6** the Hard Mechanism Change, $|\cup_{k=1}^2 \mathrm{Pa}_k(X_t^j)| > |\mathrm{Pa}(X_t^j)|$ so that $\omega_j$ always leads to a smaller size of estimated parent sets than $\hat{\omega}_j$, contrary to the definition of $\hat{\omega}_j$. Hence, $\hat{\omega}_j = \omega_j$. $\square$

With those lemmas, we can prove Theorem 1.

*Proof.* Assuming that a correct $\omega_j$ has already been obtained, from Lemma D.4 we have

$$\widehat{Pa}(X_t^j) = Pa(X_t^j), \ \forall t \geq 2\tau_{\text{ub}}, j \in [n]$$

From Lemma D.5, we know that a correct $\omega_j$ must be obtained with consistent CI tests, that is, $\hat{\omega}_j = \omega_j, \forall j \in [n]$. Therefore from Algorithm B1, we have

$$\widehat{Pa}(X_t^j) = Pa(X_t^j), \ \forall t \geq 2\tau_{\text{ub}}, j \in [n]$$

|  | $|\widehat{Pa}_1(X_t^j)|$ | $|\widehat{Pa}_2(X_t^j)|$ | $|\widehat{Pa}_3(X_t^j)|$ | $|\widehat{Pa}_4(X_t^j)|$ $\cdots$ |
|---|---|---|---|---|
| $\hat{\omega}_j = 1$ | 30 | | | |
| $\hat{\omega}_j = 2$ | 26 | 21 | | |
|  | $>$ | $>$ | | |
| $\hat{\omega}_j = 3$ | 9 | 11 | 10 | |
|  | $<$ | $<$ | $<$ | |
| $\hat{\omega}_j = 4$ | 21 | 12 | 15 | 9 |
| $\hat{\omega}_j = 5$ | 27 | 16 | 12 | 13 |
| $\cdots$ | $\cdots$ | $\cdots$ | $\cdots$ | $\cdots$ $\cdots$ |

Figure 3: In the above illustration of the "turning point," the sizes of parent sets for different estimates $\hat{\omega}_j$ are depicted as $|\widehat{Pa}_k(X_t^j)|, k \in [\hat{\omega}_j]$. It is worth noting that $\widehat{Pa}_k(X_t^j)$ represents either the true parent set or the illusory parent set of $X_t^j$. In this context, we are interested in the sizes of these parent sets. The first occurrence of the "turning point" happens at $\hat{\omega}_j = 3$ since the sizes of parent sets obtained when $\hat{\omega}_j = 2$ and $\hat{\omega}_j = 4$ are larger than the corresponding size when $\hat{\omega}_j = 3$, respectively. The term "turning point" denotes that as $\hat{\omega}_j$ increases, the size of the parent set initially decreases and then starts increasing once the local minimum is reached. The corresponding relations exist because as long as $\hat{\omega}_j$ is not a multiple of the true $\omega_j$, the estimated time partition subsets with $\hat{\omega}_j$ must be a mixture of some correct time partition subsets with $\omega_j$. Therefore, it is reasonable to use this trick rather than looking at the maximum size of the parent sets $\widehat{Pa}_k(X_t^j), k \in [\hat{\omega}_j]$ (line 17 in Algorithm B1).

If the causal mechanism is fixed across time, i.e., $\omega_j = 1, j \in [n]$, the proof of PCMCI Runge et al. [2019] showed that for all $\mathbf{X}^j \in V$,

$$X_{t-\tau}^i \to X_t^j \notin \mathcal{G} \Rightarrow X_{t-\tau}^i \to X_t^j \notin \widehat{\mathcal{G}}$$

$$X_{t-\tau}^i \to X_t^j \in \mathcal{G} \Rightarrow X_{t-\tau}^i \to X_t^j \in \widehat{\mathcal{G}}$$

Therefore $\widehat{\mathcal{G}} = \mathcal{G}$.

If $\exists \omega_j > 1$, we can simply separate the whole graph $\mathcal{G}$ into sub graphs $\{\mathcal{G}_k^{\omega_j}\}_{k \in [\omega_j]}$ consisting of only target variable $X_t^j$ with corresponding $t \in \{\Pi_k^j\}_{k \in [\omega_j]}$ and parent variables $X_{t'}^i \in Pa(X_t^j)$. Focusing only on one time partition subset $\Pi_k^j, k \in [\omega_j]$, we have

$$\widehat{\mathcal{G}}_k^{\omega_j} = \mathcal{G}_k^{\omega_j} \tag{112}$$

for any $k \in [\omega_j]$ and $j \in [n]$ based on the proof of Proposition 1 in the supplementary materials of Runge et al. [2019].

Each sub-graph $\mathcal{G}_k^{\omega_j}$ includes only variable $X_t^j$, the edges entering $X_t^j$ for time points $t \in \Pi_k^j$ and the corresponding parent variables $X_{t'}^i \in Pa(X_t^j)$. Given $\Pi^j = \underset{k \in [\omega_j]}{\cup} \Pi_k^j$ and $V = \underset{j \in [n]}{\cup} \mathbf{X}^j$, we have:

$$\widehat{\mathcal{G}} = \underset{j \in [n], \, k \in [\omega_j]}{\cup} \widehat{\mathcal{G}}_k^{\omega_j} \tag{113}$$

$$\mathcal{G} = \underset{j \in [n], \, k \in [\omega_j]}{\cup} \mathcal{G}_k^{\omega_j} \tag{114}$$

On the basis of Eq.(112), we finally have:

$$\widehat{\mathcal{G}} = \mathcal{G}$$

$\square$

# E   Turning Points

Given infinite samples, our estimate $\hat{\omega}_j$ (line 17 in Algorithm B1) is the exact value $\omega_j$ (see Lemma D.5). However, for finite samples, estimating $\omega_j$ by the equation in line 17 in Algorithm B1 does

not yield good performance when $T$ is small. While searching, larger guesses $\omega$ lead to finer time partitions in $\Pi_j$, resulting in smaller sizes for $\Pi_j^k$ (see Line 5 in Algorithm B1). Due to the power limit of CI tests on a smaller sample given by $\Pi_j^k$, the number of false negative edges increases. In order to solve this issue, we introduce *turning points*. A *turning point* is a guess $\hat{\omega}$ satisfying:

$$\max_t |\widehat{\mathrm{Pa}}_{\hat{\omega}}(X_t^j)| < \min\{\max_t |\widehat{\mathrm{Pa}}_{\hat{\omega}-1}(X_t^j)|, \max_t |\widehat{\mathrm{Pa}}_{\hat{\omega}+1}(X_t^j)|\}$$

where $|\widehat{\mathrm{Pa}}_{\hat{\omega}}(X_t^j)|$ is the estimated parent set for $X_t^j$ with periodicity guess $\hat{\omega}$. See line 19 in Algorithm B1.

We illustrate it with a special example in Fig.3. If there are several turning points, then $\hat{\omega}_j$ is the first turning point. If there is no turning point, then we obtain $\hat{\omega}_j$ using Line 17 of Algorithm B1.

The concept of the turning point is not based on any formal theorem but rather on experimental observations. In experiments, the turning point often corresponds to a multiple of the true periodicity when $T$ is not large. This occurs due to the limitations of CI tests on finite samples. In such cases, the causal graph can still be correct because the estimated time partition remains accurate. In these experiments, the accuracy rate is calculated by considering $\{N\omega_j\}_{N \in \lfloor \frac{\omega_{\mathrm{ub}}}{\omega_j} \rfloor}$ as correct estimations.

## F   Computational Complexity

Executing the PCMCI algorithm on the entire time series constitutes the initial phase of the proposed approach (Algorithm B1 line 2). The algorithm's worst-case overall computational complexity is $O(n^3\tau_{\mathrm{ub}}^2) + O(n^2\tau_{\mathrm{ub}})$, discussed in Runge et al. [2019]. Here, the symbol $n$ denotes $n$-variate time series and $\tau_{\mathrm{ub}}$ represents the upper boundary for time lags.

The subsequent computational load stemming from the remaining components of our algorithm follows a complexity of $O(\omega_{\mathrm{ub}}^2 n^2 \tau_{\mathrm{ub}})$ . This encompasses the $O(n^2\tau_{\mathrm{ub}})$ complexity associated with conducting Momentary Conditional Independence (MCI) tests on all $n$ univariate time series. The parameter $\omega_{\mathrm{ub}}^2$ here arises due to the search procedure involving $\omega$, iterating through values from 1 to $\omega_{\mathrm{ub}}$ for all $n$ univariate time series.

The runtime of the computation is further influenced by the scaling behavior of the CI test concerning the dimensionality of the conditioning set and the temporal series length $T$. For further details, see section 5.1 in Runge et al. [2019].

## G   Experiments

All experiments, including those detailed in the main paper, are conducted on a single node with one core, utilizing 512 GB of memory in the Gilbreth cluster at Purdue University.

Here, we describe how to calculate the metrics ($F_1$ score, Adjacency Precision, and Adjacency Recall) in our setting. In stationary time series, the output of the causal discovery algorithm is typically an adjacency matrix with dimensions $[n, n, \tau_{\max} + 1]$. Within the three-dimensional binary array, the value 1 signifies an edge pointing from one variable to another with a specific time lag, while 0 indicates the absence of an edge. For instance, if element $[i, j, k]$ in the matrix is 1, then there is an edge pointing from $X_{t-k}^i$ to $X_t^j$. In semi-stationary time series, due to the presence of multiple causal mechanisms, the binary edge matrix is a four-dimensional array with dimensions $[n, \Omega, n, \tau_{\max} + 1]$, where $\Omega$ is defined as Eq.(7) in the main paper. This expanded binary matrix is constructed based on the edge matrix of each variable $X_t^j, j \in [n]$, through repetition. For instance, if $\Omega = 2\omega j$, setting the third dimension of the large binary matrix to $j$ should yield $\omega_j$ potentially different parent sets (including illusory and true parent sets), each appearing twice.

We should have two such binary arrays, one representing the ground truth with dimensions $[n, \Omega, n, \tau_{\max} + 1]$ and one obtained from the algorithm with dimensions $[n, \widehat{\Omega}, n, \tau_{\max} + 1]$. If the estimator $\widehat{\Omega}$ is incorrect, those two binary arrays will have different sizes, so we can not directly compare them. To solve this problem, we do the same operation and calculate the least common multiple of $\Omega$ and $\widehat{\Omega}$. Denoting this least common multiple as $\mathrm{LCM}(\Omega, \widehat{\Omega})$, we create two four-dimensional binary arrays with dimensions of $[n, \mathrm{LCM}(\Omega, \widehat{\Omega}), n, \tau_{\max} + 1]$ based on the true edge array and the

estimated edge array, respectively, through repetition. The metrics are then computed by comparing the values in these two arrays.

## G.1 More Discussion regarding the Case Study

As stated in the main paper, we express our inability to comment on the significance of the case study results. We open a door for the related experts; if assumptions **A1-A7** are satisfied, the stationary assumption may not hold in this real-world dataset, and such periodicity exists. However, if the finding is not correct from an expert's viewpoint, the following assumptions may be violated:

- Assumption **A4** No Contemporaneous Causal Effects: There is a possibility of potential causal effects from $X_t^{\text{ta}}$ to $X_t^{\text{cp}}$ that the algorithm is unable to capture.

- Assumption **A6** Hard Mechanism Change combined with limited power of CI tests: If there is a soft mechanism change in the variables, the reliability of the CI test of two variables given their parents will be influenced by the skewed distribution of the parent variables. This effect will be exacerbated by the fact that the sample size will be shrunk by $\hat{\omega}$.

We provide a sound and robust algorithm for experts in various fields who are interested in validating the presence of periodicity within the causal mechanisms specific to their domain.

## G.2 Experiments on Continuous-valued Time Series with Exponential Noise

Considering that VARLiNGAM is a temporal extension of LiNGAM and LiNGAM is an algorithm designed for non-Gaussian data, following the work in Pamfil et al. [2020], we also construct experiments on continuous-valued time series data with Exponential noise. Shown as Fig.4(a), the performance of PCMCI$_\Omega$, PCMCI and VARLiNGAM, are quite similar with their performance on Gaussian noise. The recall rate of DYNOTEARS, however, gets worse with Exponential noise.

## G.3 Experiments on Binary Time Series

Similar to the process of generating continuous-valued time series, the generation of binary time series also involves three steps. However, the main difference lies in the last two steps. In the third step, we simulate the conditional distributions of each child variable based on all possible combinations of parent variable values. Subsequently, we randomly generate the value of the child variable by considering the corresponding conditional distribution given its parent sets.

For discrete-valued time series, a longer time length is required. To evaluate performance, we conduct a series of experiments following the same methodology as described in section 4.1. Fig.4(b) illustrates the variation in comprehensive performance with respect to $\omega_{\max}$. PCMCI$_\Omega$ demonstrates a similar performance to PCMCI in terms of the $F1$ score, indicating a well-balanced trade-off between precision and recall. This outcome is expected since discrete-valued time series demand larger sample sizes, and the increases in $\omega_{\max}$ negatively impact the power of MCI tests. This observation is further supported by Fig.5(a), where an increase in time length $T$ from 4000 to 12000 does not lead to a significant improvement in the accuracy rate of $\hat{\omega}$, while the accuracy decreases rapidly with higher values of $\omega_{\max}$.

Comparing these results to the experiments conducted on continuous-valued time series, it becomes evident that the demand for efficient samples is even more substantial for binary time series, and the influence of increasing $\omega_{\max}$ on performance becomes more pronounced.

Fig.5(b) shows how the performance of the algorithm varies across $\tau_{\max}$ and the same trade-off between recall and precision has been shown.

## G.4 More experiments on Continuous-valued time series

In this section, we conduct more experiments with continuous-valued time series with Gaussian noises.

In Fig.6(a), we test our algorithm with and without utilizing the *turning point* rule. See lines 13-14 in Algorithm B1 and section E for more information about the *turning point* rule. Let PCMCI$_\Omega$ TP

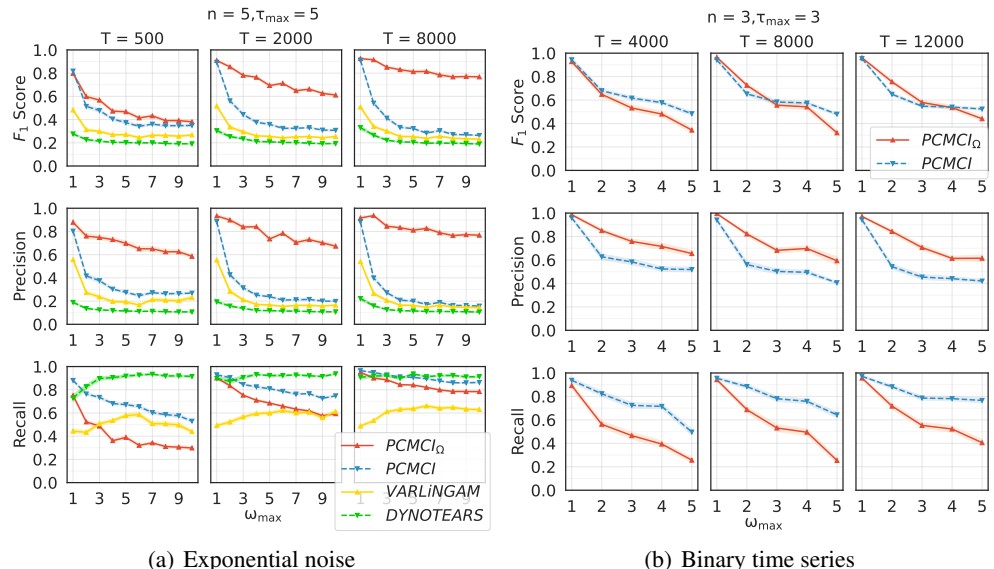

(a) Exponential noise        (b) Binary time series

Figure 4: a)$F_1$ Score, Adjacency Precision, and Adjacency Recall when $\omega_{max}$ varies for experiments on continuous-valued time series with Exponential noise, length $T = \{500, 2000, 8000\}$, $\tau_{max} = 5$ and $n = 5$. b) $F_1$ Score, Adjacency Precision, and Adjacency Recall when $\omega_{max}$ varies for experiments on binary time series with length $T = \{4000, 8000, 12000\}$, $\tau_{max} = 3$ and $n = 3$.

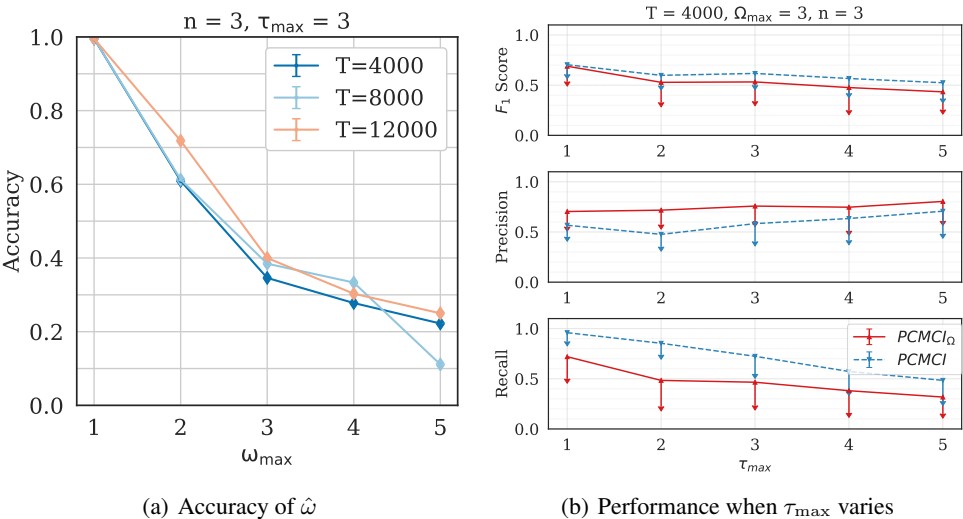

(a) Accuracy of $\hat{\omega}$        (b) Performance when $\tau_{max}$ varies

Figure 5: PCMCI$_\Omega$ is tested on 3-variate binary time series. Every marker corresponds to the average accuracy rate or average running time over 100 trials. a) The accuracy rate of $\hat{\omega}$ for different time series lengths and different $\omega_{max}$. b) $F_1$ Score, Adjacency Precision, and Adjacency Recall when $\tau_{max}$ varies for experiments with time series length $T = 4000$, $\omega_{max} = 3$ and $n = 3$.

denote the version of PCMCI$_\Omega$ that the *turning point* rule is utilized in choosing $\omega$. PCMCI$_\Omega$ non-TP means that the *turning point* rule is not applied and $\omega$ is chosen directly according to Lemma D.5.

Fig. 6(a) shows that the algorithm PCMCI$_\Omega$ non-TP and PCMCI$_\Omega$ TP have similar performance with various $T$ and $\omega_{max}$. With $T = 500$, PCMCI$_\Omega$ non-TP yields slightly larger standard errors for those metrics, compared to PCMCI$_\Omega$ TP. As time length $T$ increases, the performance of the algorithm PCMCI$_\Omega$ non-TP has consistently increased and is even slightly better than PCMCI$_\Omega$ TP.

The consistent performance of PCMCI$_\Omega$ under different chosen rules of $\omega$ supports our theoretical result; that is, the correct periodicity leads to the most sparse causal graph.

In Fig. 6(b), non-stationary time series are produced instead of semi-stationary ones. Consequently, the causal mechanisms for each univariate time series no longer appear sequentially and periodically. The proposed method performs slightly better in terms of F1 score and precision. However, the recall rate is the worst compared to other baselines.

In Fig. 6(c), we conduct experiments in the nonlinear setting. The proposed algorithms PCMCI$_\Omega$ TP and PCMCI$_\Omega$ non-TP perform the best.

In Fig. 6(d), with $\omega_{ub} < \omega_{max}$, the performance of the proposed algorithm is significantly worse compared to the scenario where $\omega_{ub} > \omega_{max}$. However, with $\omega_{ub} < \omega_{max}$, the proposed algorithm can still detect a less dense graph in comparison to other baselines. Based on these outcomes, it is essential to maintain a slightly higher $\omega_{ub}$ without significantly impacting the number of efficient samples utilized in each CI test.

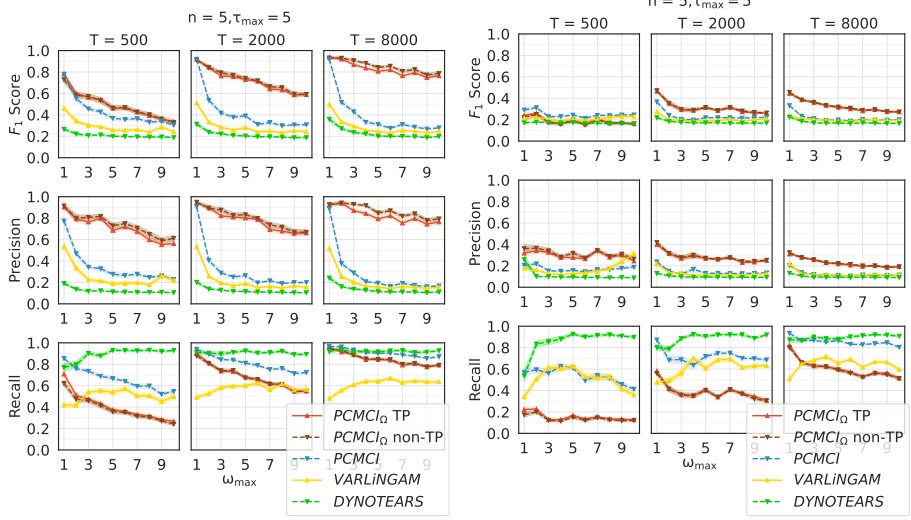

(a) Performance with and without turning point

(b) Performance in non-stationary setting without periodicity

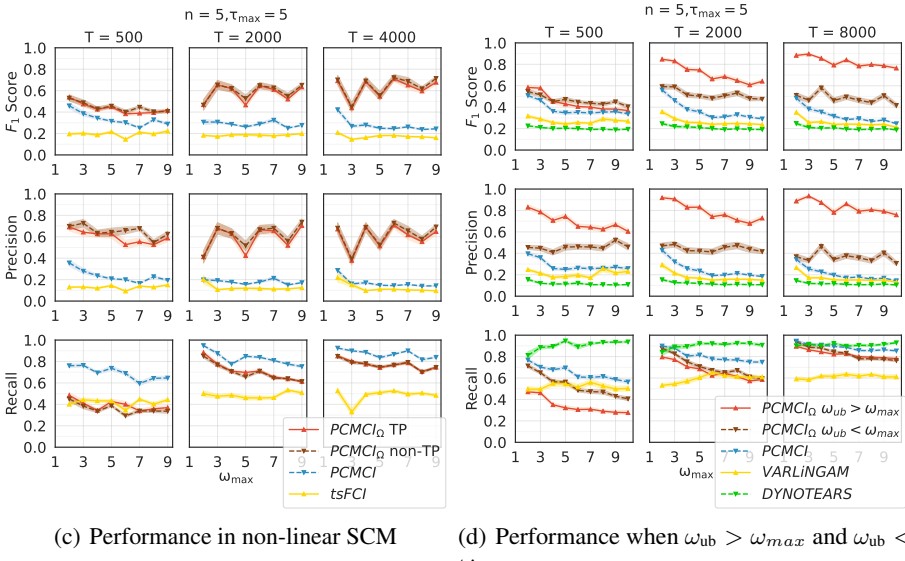

(c) Performance in non-linear SCM

(d) Performance when $\omega_{\text{ub}} > \omega_{max}$ and $\omega_{\text{ub}} < \omega_{max}$

Figure 6: Multiple algorithms are tested on 5-variate time series with different time lengths $T$. Every line corresponds to a different algorithm. Every marker corresponds to the average performance over 50 trials. In (a), the consistent performance of PCMCI under different chosen rules of $\omega$ supports our theoretical result; that is, the correct periodicity $\omega$ leads to the most sparse causal graph. In (b), data sets are in a non-stationary setting without periodicity. In (c), the structural causal model (SCM) is non-linear. In (d), algorithm PCMCI$_\Omega$ are tested under conditions that $\omega_{\text{ub}} > \omega_{\text{max}}$ and $\omega_{\text{ub}} < \omega_{\text{max}}$ respectively.

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
