# OpenReview forum: "Causal Discovery in Semi-Stationary Time Series"
_NeurIPS.cc/2023/Conference — NeurIPS 2023 poster_

### Official Review · Reviewer_ztgZ · 2023-06-25

**Soundness:** 3 good
**Presentation:** 2 fair
**Contribution:** 3 good
**Rating:** 6
**Confidence:** 4

**Summary:**

The problem of causal discovery is tackled for time series data in those cases where stationarity of multivariate time series data can not be assumed. A problem formulation based on structural causal model is given by considering the case of semi-stationary time series. Under this assumptin, the paper considers the case where a finite number of different causal mechanisms happen in a sequential manner and periodically across time. A constraint based, and non parametric, structural learning algorithm is designed and developed by leveraging on the PCMCI, described in the specialized literature. Such an algorithm is extended and adapted to cope with the considered semi-stationary setting. The causal discovery problem is tackled by standard conditional independence tests. The paper proves that the proposd algorithm is sound when asked to discover and identify causal relationships between variables of discrete time series. Numerical experiments, both from synthetic and real world data, are reported and described to comment on the performance of the proposed approach. The main contributions of the paper are in my humble opinion the following:
- designed and developing a new causal discovery algorithm for semi-stationary time series where the causal graph changes periodically.
- validation of the proposed approach through syntethic data

**Strengths:**

- the tackled problem which is of both theoretical and practical interest
- the theoretical framework of the considered problem


**Weaknesses:**

- the proposed algorithm wisely exploits the PCMCI algorithm while not being original, it looks a little as an incremental result
- numerical experiments seem to compare the proposed algorithm to algortihms which are not conceived to tackle the considered problem
- numerical experiments on real world data are disappointing. Indeed, the paper itself recognizes that not having knowledge about the considered domain the results obtained by the proposed algorithm can not be validated nor commented on.  Therefore, I ask myself which is the value of presenting such experiments?
- computational complexity is not mentioned at all
- the paper structure and organization makes it not trivial to follow it
- I found many typos

**Questions:**

- which is the coputational complexity of the algorithm you proposed?
- how does the proposed algrithm scales with respect to the number of variables of the multivariate time series?
- could you please explain why the following works had not been taken into account? maybe they are slighlty out of the scope but still linked to the problem you tackled (Learning Continuous Time Bayesian Networks in Non-stationary Domains), (On-homogeneous  dynamic bayesian networks  with  bayesian  regularization  for  inferring  gene  regulatory  networks  with gradually time-varying structure) (Learning non-stationary dynamic bayesian networks)
- line 104: could you please explain the meaning of setting P(V) different from 0?
- line 105: the notation used for the pair of variables is not properly coherent, indeed Xj is a univariate timeseries contained in V, which has not been defined. Furthermore, what do you mean by variables here? two time serires?
- line 115: what do you mean by finite maximum lag?
- would you be that kind to comment explicitly and not only formally on the assumptions made about the time when nonstationarity occurs, i.e., the casual graph changes its structure? synchronous? asynchronous for each time series? could you discusse more on this aspect? what about parameters non stationarity? I guess you should consider this is excluded to happen, or not?
- line 156: the definition of n is a little confusing. Indeed, n belongs to a set that is defined as a function of n again, would you be that kind to help me understand?
- formula (14), the expernal p is the same probability ad the p inside brackets?
- line 243, I read "continuous time series in four steps", but I found only three of them
- would you also consider the structrual hamming distance to measure the performance of the proposed algorithm



**Limitations:**

- computational complexity of the proposed is not mentioned at all
- comparison has not been performed with algorithm natively designed for semi-stationary time series data. Maybe no such algorithms exist?
- results on real world data are of no help because no ground thruth is available and no domain expert comments are given to corroborate it

---

> ### Author Rebuttal · Authors · 2023-08-10
>
> We really appreciate the feedback. Regarding your valuable questions, please see below for our responses.
>
> ## Q1. Computational Complexity
> Executing the PCMCI algorithm on the entire time series constitutes the initial phase of the proposed approach (designated as "line 1" in **Algorithm 1**). The algorithm's worst-case overall computational complexity is delineated by $O(n^{3}\tau_{ub}^{2})$. Here, the symbol $n$ denotes the multivariate nature of the temporal data, and $\tau_{ub}$ represents the upper boundary for time lags.
>
> The subsequent computational load stemming from the remaining components of our algorithm follows a complexity of $O(\omega_{ub}^2n^{2}\tau_{ub})$. This encompasses the $O(n^{2}\tau_{ub})$ complexity associated with conducting Momentary Conditional Independence (MCI) tests on all $n$ univariate time series. The parameter $\omega_{ub}^2$ here arises due to the search procedure involving $\omega_{j}$, iterating through values from $1$ to $\omega_{ub}$ for all $n$ univariate time series.
>
> The runtime of the computation is further influenced by the scaling behavior of the conditional independence test concerning the dimensionality of the conditioning set and the temporal series length $T$. For further details, see section 5.1 in the work by Runge et al.[2019].
> ## Q2. Novelty
> Thank you for your comment.  We agree that our solution leverages PCMCI and builds on it. However, we believe that the novelty of our approach lies in simplicity by carefully leveraging it using algorithmic ideas. Please note that prior to our work, it was not clear if we needed to rethink the causal discovery problem from scratch for non-stationary settings or if some of the existing work could be leveraged in a clever way. Our algorithmic contribution demonstrates the latter.
> ## Q3.  Lack of appropriate Baselines
> We agree with the reviewer. To the best of our knowledge, there are no non-parametric baselines that can handle non-stationarity or semi-stationarity in the data with a window causal graph.
> ## Q4. Experiment in the case study
>
> While we understand that our results are based on simulated data, it is fairly common in the area to benchmark against the simulated settings that we considered.
> As one of the main contributions of our work is the introduction of the problem, we believe that our empirical contributions along with the problem formulation are significant.  Moreover, many results in causal discovery and inference rely on synthetic simulations, which is necessary in the absence of ground truth.
>
> ## Q5. Related Literature
> We appreciate the valuable literature you've shared; it bears significant relevance to our ongoing project. Regarding our proposed algorithm here, we emphasize a non-parametric approach capable of uncovering causal relationships with time-lag effects and adapting to both abrupt changes in causal mechanisms at each time point while also addressing consistent causal mechanisms as long as periodicity is present. In contrast, the works mentioned are not distribution-free and generate summary causal graphs as outputs.
>
> ## Q6. $P(V)\neq 0$
> The probability of any given realization involving all variables within $V$ should not be zero. For instance, consider $V=\\{X,Y\\}$; when computing $P(X|Y)=\frac{P(X,Y)}{P(Y)}$, we aim to avoid scenarios where the denominator or numerator becomes zero. This is needed in the proof, and this prerequisite is already met by **Assumption A7**. $P(V) \neq 0$ is also called the Positivity assumption.
>
> ## Q7. Clarification on some notations in Line 105
> The notation $X, Y\in V$ signifies that any two random variables, denoted by $X$ and $Y$, belong to the set $V$.  $X, Y\in \mathbb{R}^{1}$. Meanwhile, $S$ represents a subset of $V$. As indicated in line 101, $X_{t}^{j} \in \mathbb{R}^{1}$ designates an individual variable, whereas $\mathbf{X}^{j}_{t\in[T]} \in \mathbb{R}^{T}$ represents a time series.
> ## Q8. Finite maximum lag
> This implies that as $t$ approaches infinity, $X^i_1$ will not serve as a cause for $X^j_t$. The maximum time lag $\tau_{\text{max}}$, as defined in Eq.1, must remain finite.
> ## Q9. Non-stationarity in the Semi-Stationary setting
> Each univariate time series $\mathbf{X}^j$ in the n-variate time series $V$ can have its own periodicity $\omega_j, j\in[n]$ in the causal mechanisms. Within each cycle of $\omega_j$ time points, the same causal mechanism is reiterated, resulting in $\omega_j$ distinct causal mechanisms for the time series $\mathbf{X}^j$.
>
> In accordance with **Assumption A6**, when the causal mechanisms governing two variables, $X^j_{t1}$ and $X^j_{t2}$, have undergone alteration, it follows that their respective parent sets cannot remain identical. In essence, when the parameters within the structural causal model shift, there is a corresponding change in the parent set as well.
> ## Q10. Line 156, definition of $n$
> $n$ belongs to a subset of $\mathbb{N}$ that should satisfying $\tau_{max}+q-1+n\delta \leq T$, i.e, $n\leq (T-(\tau_{max}+q-1))/\delta$
> ## Q11. Formula 14, two $p$
> Yes, they both denote the probability. That means the chance of the two probabilities enclosed within the brackets being identical is zero.
> ## Q12. Line 243 typo
> There should be only three steps.
> ## Q13. Structrual hamming distance
> We did not use structural hamming distance because of the semi-stationary context, wherein the estimated binary edge array's dimensionality grows with rising estimated $\hat\omega_{j}$. As a result, larger $\hat\omega_{j}$ values inherently amplify the dissimilarity in SHD between the true and estimated edge arrays. This complexity makes direct comparisons challenging. Precision, recall, and F1 score, on the other hand, are intrinsically normalized within the range of [0,1], rendering them suitable metrics for the semi-stationary context.

---

> > ### Comment · Reviewer_ztgZ · 2023-08-11
> > **Read your rebuttal**
> >
> > Thank you, I went through your rebuttal question by question or lets say answer by answer.
> > I found many answers being very useful to clarify aspects that in the original submission were not that clear to me.
> > However, I'm still not satisfied by answers to lack of baselines, I understand the author's point but I would have expected a more involved discussion on this aspect. Furthermore, if you describe experiments on real data where you do not have ground truth and you do not have knowledge about the domain I still miss how you can validate your findings. Then, also the answer to Q6 is puzzling, indeed if you consider continuous variables you always have that the variable X takes on a specific continuous value x with probability zero, so I miss the point.

---

> > > ### Author Response · Authors · 2023-08-12
> > >
> > > Thank you for your insightful comments and valuable questions.
> > >
> > > **Baselines**: Sorry we didn't write this before due to the limited space. Here is our detailed discussion.
> > >
> > > Indeed, several baseline methods are designed to handle non-stationary temporal data, such as CD-NOD[1] and JIT-LiNGAM[2]. However, it's important to note that the outputs of these baselines are **summary graphs**, allowing them to identify the causal relationship between time series $X^j$ and $X^i$ without pinpointing the specific time lag $\tau$. In contrast, our method enables the detection of the precise variable $X^j_{t-\tau} \in X^j$ that causes the behavior in $X^i_t$, presenting a distinguishing feature. These approaches highlight distinct aspects from ours. While summary graphs offer a degree of insight, they are inherently less informative about the underlying causal structure in our specific case compared to the detailed window graphs produced by our method.
> > >
> > > To showcase our method's capacity to handle periodicity in causal mechanisms with time-lag effect, a capability lacking in other approaches, we will introduce a related baseline called "Regime-PCMCI" in our new experiment. It's worth noting that this algorithm necessitates a linear model (while our method is non-parametric) and additionally mandates precise foreknowledge of the number of causal mechanisms inherent within the time series. Remarkably, based on several experiment results we have, our algorithm's superior performance is sustained even when the accurate count of causal mechanisms is supplied to the "Regime-PCMCI" method. Given the considerable time requirement of the "Regime-PCMCI" method, we regret to say that the result plot is currently unavailable. However, we intend to incorporate the said plot into the paper later.
> > >
> > > The significance of the comparison depicted in Fig. 3(a) with the baselines we have is apparent when considering the following points:
> > >
> > > 1. Our algorithm excels in the reduction of False Positive edges, as indicated by the higher Precision, particularly when compared to the stationary algorithm employed in a semi-stationary setting. This outcome effectively validates **Lemma 3.4**, providing a solid foundation for our approach.
> > >
> > > 2. With the expansion of the sample size, a notable reduction in False Negative edges becomes evident, as evidenced by the recall rate approaching that of PCMCI, which stands out as the best-performing among the baseline methods.
> > >
> > > 3. In cases where the ground truth involves periodic causal mechanisms, the misuse of a stationary algorithm leads to a significant amount of error, emphasizing the critical importance of employing an appropriate method, such as ours, to account for such complexities.
> > >
> > > As far as we know, our method is the first to address the semi-stationary Structural Causal Model (SCM) with the capability to accommodate periodicity. We hope that our approach can establish itself as a potential reference for individuals interested in periodic causal mechanisms.
> > >
> > > **Case Study**:
> > > Thank you for the comment. We will add the following discussion in the camera-ready paper to avoid potential misunderstanding regarding our conclusion in the case study.
> > >
> > > We could not comment on whether the result of the case study is significant. We open a door for the related experts; if assumptions A1-A8 are satisfied, the stationary assumption may not hold in this real-world data, and such periodicity exists. However, if the finding is not correct from an expert's viewpoint, the following assumptions may be violated:
> > >
> > > 1. **Hard Mechanism Change** combined with limited power of CI tests: if there is a soft mechanism change in the dataset, the reliability of the CI test of two variables given their parents will be influenced by the skewed distribution of the parent variables. This effect will be exacerbated by the fact that the sample size will be shrunk by $\omega$.
> > >
> > > 2. **No Contemporaneous Causal Effects**: There's a possibility of potential causal effects from $X^{ta}_t$ to $X^{cp}_t$ that we're unable to capture in our analysis under this assumption.
> > >
> > > Our method provides a sound and robust (shown as figures in the Rebutall pdf file) algorithm for experts in various fields who are interested in validating the presence of periodicity within the causal mechanisms specific to their domain.
> > >
> > > **More details about $p(V)\neq 0$**: Suppose Y is a continuous random variable. p(Y) here is the probability density, which is always positive in the domain of Y. For example, this would be a case for any non-degenerate jointly Gaussian distribution.
> > >
> > > Thank you again and we wholeheartedly invite further discussions.
> > >
> > > [1]Huang, Biwei, et al. "Causal discovery from heterogeneous/nonstationary data." The Journal of Machine Learning Research 21.1 (2020): 3482-3534.
> > >
> > > [2]Fujiwara, Daigo, et al. "Causal Discovery for Non-stationary Non-linear Time Series Data Using Just-In-Time Modeling." 2nd Conference on Causal Learning and Reasoning. 2023.

---

> > > > ### Comment · Reviewer_ztgZ · 2023-08-12
> > > > **yes but in the rebuttal you wrote P(X) and not p(X), but now that you specified, all is clear**
> > > >
> > > > yes but in the rebuttal you wrote P(X) and not p(X), but now that you specified, all is clear

---

> > > > > ### Author Response · Authors · 2023-08-14
> > > > >
> > > > > We sincerely appreciate your dedication and time. We wholeheartedly welcome further discussions.
> > > > >
> > > > > Should our rebuttal effectively address the concerns, we kindly hope you consider it for acceptance.

---

> > > > > > ### Comment · Reviewer_ztgZ · 2023-08-16
> > > > > > **Thank you, you almost soled all my residual doubts and so ...**
> > > > > >
> > > > > > I'm gonna increase my rating of your paper, also after having read opinion from other colleagues and subsequent discussion.

---

> > > > > > > ### Author Response · Authors · 2023-08-16
> > > > > > >
> > > > > > > Your commitment to reviewing our submission and sharing invaluable insights is deeply appreciated. Thank you for investing your time and effort.
> > > > > > > We also want to express our gratitude for the reconsideration.

---

### Official Review · Reviewer_ZpAB · 2023-06-26

**Soundness:** 4 excellent
**Presentation:** 3 good
**Contribution:** 3 good
**Rating:** 6
**Confidence:** 3

**Summary:**

The paper describes a constraint-based causal discovery method for semi-stationary SCMs, where  the causal structure is periodically changing over time.
This type of structure is relevant in real-life scenarios where seasonal or diurnal variation is present for example.
The problem is rigorously formalized and and algorithm is give to reconstruct the SCM. As a first step the algorithm identifies a superset of parents using PCMCI naively,
then searches for the correct periodicity for all variables by minimizing parent set size. The author provides theoretical guarantees that the reconstructed causal graph
is indeed the generating graph in the infinite sample size limit, if all assumptions hold.

**Strengths:**

The paper describes a novel extension of PCMCI for semi-stationary SCMs, and gives theoretical guarantees. The probrem formulation is sufficiently rigorous, and the algorithm is described in sufficient details. This makes the works reproducible.

The method relaxes a quite frequent assumption of causal discovery methods in the temporal domain: stationarity. Even if this new relaxed assumption also likely violated by many real life dataset, it is clearly an improvement  in cases when stationarity is clearly a wrong assumption.

**Weaknesses:**

The main limitation I see is the following: For example, if the causal effect changes by the day of the week but the samples are collected in every hour, this periodicity is hard to represent in the presented framework. Given the process inherent timescale is hourly, we cannot just subsample the data as this would induce transitive dependencies. However in the current form the method will try to fit a new causal mechanism in every hour (until the period is reached of course).
It would be good to know what happens if this type of data was fed to the algorithm, as this is hard to avoid. To avoid the identification of the transitive dependencies as genuine we are motivated to work as fine timescale as available.

Similar question applies for the "sudden" mechanism change. In real life most likely the sudden change will transform to a smooth change as the sampling frequency increases (for example seasonal effect in a daily scale). Most likely the effect of the Atlantic air temperature on the Central Pacific air temperature does not disappears suddenly.

**Questions:**

l257: **Following the previous work in Huang et al. [2020], F1 score, Adjacency Precision, and Adjacency Recall are used to measure the performance**
Please specify in details how you compute these metrics. Classical methods reconstruct a single causal mechanism per variable, while the ground truth has multiple.

l259: **The standard error of the averaged statistics, displayed either by color filling or by error bars is usually too small to be observed.**
You mean here the standard error of the mean estimate? You can use the sample variance instead.

Fig 2a, In horizontal axis $\omega_{max}$ is shown. Has the code been executed with the same $\omega_{ub}$ for all cases, or the $\omega_{ub}$ was smaller for smaller $\omega_{max}$ trials?
In general what hyper parameters (upper bounds, significance thresholds) were selected? What CI test was used?

In the Case Study, if you have monthly data, and Central Pacific has 3 parent sets, it seems to me the causal effect is different in every month say the time partitions are {January,April, July, .. } {February,May, August, ..} {March, June, September, ...}, how you come to the conclusion that: **"the causal effect from the tropical Atlantic air temperature to the Central Pacific air temperature would disappear every quarter of a year"*** ? I am not sure I understand this point.

Minor comment:
I would suggest to use "discrete valued" and "continuous valued" terminology instead of "discrete time series" / "continuous time series", as it can be mistaken with discrete and continuous time.

How the  sampling frequency influences the reconstruction of the SCM? (See weaknesses part.)

**Rebuttal**: I have read the author's rebuttal and we exchanged further comments. I still see the paper having potential interest for the audience of this conference. I had one main open question, the result of the case study.  Its result is unfortunately quite implausible, therefore I cannot further increase my score.

**Limitations:**

The author points out some limitation of the method. For example the higher sample complexity of the method relative to vanilla PCMCI.
It is not clear how realistic the present semi-stationarity assumption.

---

> ### Author Rebuttal · Authors · 2023-08-10
>
> We really appreciate the feedback. Regarding your questions, please see below for our responses.
>
> ## Q1. Sampling issue
>
> Thank you for this great question.
>
> Our proposed method can handle the "unchanged" causal mechanism as long as there exists a periodicity in how the causal mechanisms change over time.
>
> For instance, if the causal effect changes by the end of the day but the samples are collected in every hour, considering each number as representing a distinct causal mechanism, the causal mechanism progression of $\mathbf{X}^{j}$ over time could be [1,1,...,1,2,2,...,2,3,3,...,3,4,4,...,4,...,7,7,...,7,1,1,...,1,2,2,...,2,3,3,...,3,...4,4,...,4,...,7,7,...,7,...]. Even though the causal mechanism doesn't change every time point (every hour for example), the causal mechanism will eventually repeat itself every week and the algorithm will treat the sequence [1,1,...,1,2,2,...,2,3,3,...,3,4,4,...,4,...,7,7,...,7] for one week as a single unit, eventually leading to a periodicity of 24*7=168 hours=7 days.
>
> In this situation, our proposed algorithm will have 168 correspondingly distinct time point subsets where the 1th, 169th, and 337th causal mechanism "1" will be included in the same time point subsets instead of having all causal mechanism "1" in the same time point subsets like [1,2,3,4,5,6,7,1,2,3,4,5,6,7,...,1,2,3,4,5,6,7]. Hence, the drawback is that the effective sample used in each MCI test for the first case will be 1/24 of the samples used for the second case. That is, the sample size used in MCI tests is shrunk.
>
> In our ongoing research, our goal is to overcome this limitation and use the samples more efficiently in the first case.
>
> ## Q2. "sudden" change
>
> As for the "sudden" change in the causal mechanism, we can handle the "soft" change as long as the periodicity of the "soft" change mechanism is not a multiple of the periodicity of the "hard" change mechanism.
>
> For instance, suppose the causal mechanism progression of $\textbf{X}^{j}$ over time is [1,2,3,1,2,3,1,2,3,...] and the change between causal mechanism "1" and "2" is "soft". More specifically, suppose the only incoming edge of $X^j_t$ in causal mechanism "1" and "2" is $X^i_{t-1}$, but the causal effect between $X^i_{t-1}$ and $X^j_t$ in causal mechanism "1" is stronger than that in causal mechanism "2". This is a violation of the **Assumption A5** Hard Mechanism Change.
>
> Suppose in causal mechanism "3", **Assumption A5**  still holds. Our algorithm will treat this case as the same as [1,1,3,1,1,3,1,1,3,...]. Because of the causal mechanism "3", we will still have a periodicity of 3, and the time partition will still be correct. Even though there are no sudden changes between causal mechanisms "1" and "2", the corresponding samples will still be partitioned according to their causal mechanism, and then the causal effect will be estimated correctly.
>
> However, without causal mechanism "3", the algorithm will fail.
>
> On the other hand, if the periodicity in the "soft" change mechanisms is 2 with [1,2,1,2,...], then the periodicity of the "hard" mechanism change is 1 with [1,1,1,1,...], therefore the algorithm will tell us that the periodicity is 1, instead of 2. Hence, the algorithm can handle the "soft" change as long as the periodicity of the "soft" change mechanism is not a multiple of the periodicity of the "hard" change mechanism.
>
> Thank you for this great question that inspires us to relax the **Assumption A5** to handle "soft" change in some cases. We will update this in the camera-ready version.
>
>
> ## Q3: The metrics.
>
> Thank you for the comment, and we will include the explanation in the paper later.
>
> During our metric evaluation process, we will compute the least common multiple of $\Omega$ and $\hat{\Omega}$, where $\Omega$ itself represents the least common multiple of all $\omega_{j}, j\in[n]$, as defined in Eq. 7. Designating this least common multiple as "lcm," we construct two binary arrays of edges with dimensions $[N, lcm, N, \tau_{max}+1]$ for both the true causal graph and the estimated graph. In these arrays, the value 1 signifies an edge connecting one variable to another with a specific time lag, while 0 indicates the absence of an edge. The metrics are subsequently computed based on the difference between these two arrays of edges.
>
> ## Q4: Standard error
>
> We appreciate your suggestion, and we will make the necessary updates accordingly. To maintain consistency, we will continue to utilize the standard error for the new experiments. These updates will be synchronized with the previous experiments in the camera-ready version.
>
> ## Q5: hyperparameters and CI tests.
>
> We thank you for your comment, and we will incorporate these specific details into the main paper.
>
> For the continuous-valued time series, $\omega_{ub}=15$ for all cases no matter what $\omega_{max}$ is, and $\tau_{ub}=20$ for all cases. CI test used in continuous valued time series is a partial correlation test. For the discrete valued time series, $\omega_{ub}=7$ and $\tau_{ub}=7$ for all cases. CI test here is a conditional mutual information test.
>
> ## Q6: Conclusion in the case study
>
> The casual edge from $X^{ta}_{t-1} \text{to} X^{cp}_t$ disappears in Jan, April, June, and October, while in the remaining months, the edge exists. Therefore we have an initial conclusion that the causal effect of tropical Atlantic air temperature on the Central Pacific air temperature disappears every quarter of a year.
>
>
> ## Q7: "discrete valued" and "continuous valued" time series
>
> We greatly appreciate your supportive comment, and we will revise the terminology accordingly to prevent any potential misunderstandings.

---

> > ### Comment · Reviewer_ZpAB · 2023-08-11
> > **Reply to Rebuttal**
> >
> > I would like to thank the authors for their time to answering my (and other reviewers') questions.
> >
> > I am mostly satisfied with the answers and explanations provided. I only mention here the questions where I have some additional comment or follow up questions.
> >
> >
> > **Q1:** Indeed, my worry was exactly the diminishing sample size per CI test.
> >
> > **Q3:** So I assume if a method cannot handle non-stationarity ($\hat{\Omega} =1$), the reconstructed mechanism is repeated $\Omega$ times and compared to the ground truth. Am I correct?
> >
> > **Q5:** I misunderstood this in the original paper but indeed it was a mistake from my side. However, now I see that indeed this finding is highly implausible. Are you aware of any assumption violation that would produce this kind of artifact ?
> >
> > The provided new results in the PDF demonstrate that the method also seem to have a level of robustness on datasets where the assumptions are violated.

---

> > > ### Author Response · Authors · 2023-08-11
> > >
> > > We appreciate your insightful comments, which are truly inspiring.
> > >
> > > **A1**: We recognize that there's room for more effective sample utilization in the real-world application of our algorithm when dealing with periodic causal mechanisms that allow for the repetition of the same causal mechanism.
> > >
> > > When there are only a few instances of the same causal mechanism, such as 2, reducing the samples per CI test by half might be reasonable in certain scenarios. However, if there's a substantial number of repeated causal mechanisms, such as 200 instances, it may be more appropriate to treat this issue as a change point detection problem, which is currently a topic of our ongoing research. If the repetition falls somewhere between being small and significantly large, we don't have a definitive solution at this point, but it presents an intriguing open question worth exploring.
> > >
> > > Your insightful question is greatly appreciated, and we intend to incorporate it into the discussion of limitations within our paper.
> > >
> > > **A3**: Yes.
> > >
> > > **A6**: We deeply appreciate your valuable input on the case study. We claimed in the paper that the significance of these results is under-explored, making your comment essential. There are specific assumptions that may be violated, as follows:
> > >
> > > **Hard Mechanism Change combined with the content in Q1**: If a soft mechanism change occurs in $X_{t-1}^{cp}$, the reliability of the CI test between $X_{t-1}^{ta}$ and $X_t^{cp}$ given $X_{t-1}^{cp}$, will be impacted by the skewed distribution of $X_{t-1}^{cp}$. It is further impacted by the fact that the sample size here is 900/3=300.
> > >
> > > **No Contemporaneous Causal Effects**: There's a possibility of potential causal effects from $X^{ta}_t$ to $X^{cp}_t$ at the same time point $t$ that we're unable to capture in our analysis.
> > >
> > > Thank you for raising these crucial points; we will include these comments in our case study section.
> > >
> > > In conclusion, we wholeheartedly invite further discussions.

---

### Official Review · Reviewer_YnE6 · 2023-07-05

**Soundness:** 3 good
**Presentation:** 2 fair
**Contribution:** 3 good
**Rating:** 5
**Confidence:** 3

**Summary:**

## Summary
The paper addresses the problem of discovering causal relations in semi-stationary time series data, where a finite number of causal mechanisms occur sequentially and periodically across time. The authors propose a constraint-based, non-parametric algorithm called $PCMCI_\Omega$ to handle this type of time series data.

The main contributions are
1. The proposed algorithm, PCMCIΩ, is an extension of the PCMCI algorithm designed for stationary time series data. It leverages the PCMCI algorithm to systematically discover the superset
and use CI test to find out the correct periodicity and remove unnecessary parents.

2. The soundness of the PCMCIΩ algorithm is demonstrated through a theoretical analysis, showing that it can recover the true causal graph under specific assumptions.

3. The authors validate the effectiveness of the PCMCIΩ algorithm through experiments on both continuous and discrete synthetic time series data. Although I think the baselines can be stronger.



**Strengths:**

Strengths of the paper:
1. Originality and significance: The paper proposes a novel algorithm, PCMCIΩ, to address the challenging problem of causal discovery in semi-stationary time series data. Fewer work have considered such settings and most of the previous method assume stationarity. This is a step towards generalising causal discovery to non-stationary time series.

2. Soundness: This work provides a theoretical guarantees on the soundness of the proposed method, and validate it with experiments.


**Weaknesses:**

The main weakness of this paper are
 1. Limited scope of experiments: Although the authors have performed experiments on synthetic continuous and discrete time series data, as well as a real-world climate dataset. However, the baseline selection is too simple. For example, VARLinGaM and DYNOTEARS are both linear models with stationarity assumptions. It is expected to outperform those two methods. I suggests including more stronger baselines, such as Granger causality baselines [1], and SCM-based method [2].

2. The paper is hard to understand when I first read it. There are many definitions defined in section 2, without knowing why they are important. The graphical illustration helps a little bit, but still confuses me on why we have to define them. For example, I am not sure I understand why Illusory Parent sets requires a subscript. Like why we need $Pa_2(X_7^1)$ and $Pa_3(X_7^1)$. Isn't it the same as $Pa_1(X_8^1)$? Since the following theoretical analysis extensively uses these definitions, I recommend the author to be very clear on these definitions and should make them easier to understand for the readers.

3. How does your method performs when there are mismatches in the assumptions? For example, the ground-truth is stationary/non-stationary. When the lag upper bound and period upper bound are shorter than the truth. etc. More ablation study would be helpful.

4. What are the potential limitations of the proposed method? Some discussion should be added. For example, instantaneous effect can be important when aggregation happens. Computational complexity with dimensionality, etc.

[1] Khanna, S., & Tan, V. Y. (2019). Economy statistical recurrent units for inferring nonlinear granger causality. arXiv preprint arXiv:1911.09879.

[2] Gong, W., Jennings, J., Zhang, C., & Pawlowski, N. (2022). Rhino: Deep causal temporal relationship learning with history-dependent noise. arXiv preprint arXiv:2210.14706.


**Questions:**

1. I am a bit confused on line 123, why minimum periodicity has the same symbol as the number of causal mechanisms in $V$?

2. In definition 2.3, eq.8, the subscript $k$ is not showing in the latter equations. What does this k means?

3. Can you make the definition of Illusory parent sets easier to understand and explain the intuition behind it?

4. For figure 1, why $Z_2^1$ is identical to $Z_1^7$? I thought $Z_2$ defines the Markov chain for second time series? Also, you use superscript in $X$ to indicate time series number, I think $Z$ should follow the same pattern to avoid confusion.


**Limitations:**

The author does not explicitly provide a discussion on the limitations. Hence, I suggest the author to include a limitation section.

---

> ### Author Rebuttal · Authors · 2023-08-10
>
> Your insights are greatly appreciated as we work towards enhancing the robustness of our method. Regarding your questions,
>
> ## Q1. Scope of experiments
>
> To our best knowledge, there is no such non-parametric algorithm that can handle the periodicity in causal mechanisms and discover the time-lag effect with a window causal graph.
>
> For the Granger causality baselines [1], the estimated causal graph is a summary causal graph, while the output of our proposed algorithm is an estimated window causal graph. They highlight distinct facets. Summary graphs are less informative than the window graphs about the underlying causal structure in our case.
>
> For the SCM-based method [2], we have contacted the authors and will conduct the experiments as soon as we get the code.
>
> Based on your valuable suggestions, we conduct more experiments in the nonlinear setting with the baseline method "tsFCI." In the new experiments, shown as **Fig.1(b)** in the Rebuttal pdf file, the SCM are non-linear and the proposed algorithm performs well.
>
> As for the "non-stationary" assumption, to showcase our method's capacity to handle periodicity in causal mechanisms with time-lag effect, a capability lacking in other approaches, we will introduce a related baseline called "Regime-PCMCI" in our new experiment. It's worth noting that this algorithm necessitates a linear model and additionally mandates precise foreknowledge of the number of causal mechanisms inherent within the time series. Remarkably, based on several experiment results we have, our algorithm's superior performance is sustained even when the accurate count of causal mechanisms is supplied to the "Regime-PCMCI" method. Given the considerable time requirement associated with the "Regime-PCMCI" method, we regret to say that the result plot is currently unavailable. However, we intend to incorporate the said plot into the paper at a later stage.
>
> ## Q2. Visualization of definitions
>
> We appreciate your suggestion, and we will incorporate additional figures that illustrate each definition separately into the supplementary material.
>
> ## Q3. Ablation study
> We have conducted new experiments for the ablation study.
>
> Our proposed algorithm can handle stationary SCM as stationary SCM is a special case of semi-stationary SCM where $\Omega=1$.
>
> In a non-stationary setting without periodicity, shown as **Fig.1(c)** in the Rebuttal pdf file, the proposed method performs slightly better in terms of F1 score and precision. However, the recall rate is the worst.
>
> If $\omega_{ub}<\omega_{max}$, the performance of the proposed algorithm decreases, but still can detect a sparser graph, shown as **Fig.1(d)** in the pdf file.
>
> ## Q4. Limitations
>
> **Assumption A4** No Contemporaneous Causal Effects is needed in the underlying Markov chains of the time series. See line 39 in the Appendix.
>
> The worst case complexity of our algorithm is $O(n^{3}\tau_{ub}^{2})$+$O(\omega_{ub}^2n^{2}\tau_{ub})$. The runtime of the computation is further influenced by the dimensionality of the conditioning set in CI tests. This set's maximal size is $2+|Pa(X^j_t)|+|Pa(X^i_{t-\tau})|$, coupled with the temporal series length $T$. Refer to section 5.1 in the work by Runge et al.[2019] for more details.
>
> ## Q5. Minimum Periodicity is the number of causal mechanisms.
> As defined in Eq.4 in **Definition 2.2**, the causal mechanisms of $\mathbf{X}^{j}$ follow a sequential and periodic pattern, occurring every $\omega$ time points. The smallest value of $\omega$ that satisfies this condition is termed the periodicity of $X^{j}$, which necessitates $\omega$ causal mechanisms.
>
> To illustrate, considering each number as representing a distinct causal mechanism, the causal mechanism progression of $\mathbf{X}^{j}$ over time could be [1,2,3,1,2,3,1,2,3,...]. In this instance, $\mathbf{X}^{j}$ comprises 3 distinct causal mechanisms, which cyclically repeat every 3,or 6, or 9,..., or $3N,N\in \mathbf{N}^{+}$ time points. The smallest value of $3N$ here is 3, which is the number of causal mechanisms.
>
> However, it's important to note that these $\omega$ causal mechanisms need not to be entirely unique. For instance, the causal mechanism progression of $\mathbf{X}^{j}$ over time could be [1,1,2,1,1,2,1,1,2,...]. In this scenario, the algorithm treats the sequence [1,1,2] as a single unit, effectively resulting in a periodicity of 3 while containing only 2 distinct causal mechanisms. To maintain simplicity, we still refer to this as having 3 causal mechanisms.
> ## Q6. Updated Definition 2.3 *Time Partition*
> The updated definition should be as follows.
>
> A time partition $\Pi^{j}(T)$ of a univariate time series $X^{j}$ in a Semi-Stationary SCM with periodicity $\omega_{j}$ is a way of dividing all time points $t\in[T]$ into a collection of non-overlapping non-empty subsets $\Pi^{j}_{k} (T),k\in [\omega_j]$ such that
>
> $\Pi^j_k(T):=\\{t:\tau_{\max}+1 \leq t \leq T, (t \bmod \omega_j)+1=k \\} $
>
> where $\bmod$ denotes the modulo operation. For instance, $5\bmod3=2$.
>
> In this context, by gathering all time points $t$ where the corresponding variable $X^j_t$ share the same causal mechanism, we can form $\omega_{j}$ distinct subsets of time points, denoted as $\Pi^j_k(T)$, where $k$ ranges from 1 to $\omega_{j}$.
>
> ## Q7. Markov chains
> The Discrete-time Markov Chains $\\{Z^q_n\\}$, where $q\in [\delta]$ denotes the index of Markov Chains and $n$ denotes the state index of each Markov Chain, are defined for the entire multivariate time series $V$ and are not defined for individual univariate components $\textbf{X}^{j}\in V$. When a specific value of $q\in [\delta]$ is chosen, the sequence $\\{Z^q_n\\}_\{n\in \mathcal{N}\}$ in **Definition 2.5** constitutes a Markov Chain, with $n$ serving as the progressing index.
>
> In Fig.1, there is no $Z^7_1$ because there are only six Markov Chains associated with $V$, and $Z^1_2$ denotes the second state within the first Markov Chain.

---

> ### Comment · Reviewer_YnE6 · 2023-08-16
>
> Thanks for the additional experiments provided by the authors. They managed to address my concerns, and hope the revised version to have a more clear presentation. I will keep my current score.

---

> > ### Author Response · Authors · 2023-08-16
> >
> > We appreciate your feedback, and we're pleased that the experimental results, guided by your insightful suggestions, have effectively addressed your concerns. We are going to refine the presentation to enhance the clarity of definitions in the forthcoming version.
> >
> > Thank you once again!

---

### Official Review · Reviewer_nsZ6 · 2023-07-07

**Soundness:** 2 fair
**Presentation:** 2 fair
**Contribution:** 2 fair
**Rating:** 5
**Confidence:** 3

**Summary:**

The authors propose a causal discovery algorithm for a class of periodic random processes called semi-stationary process. The algorithm is built upon an existing algorithm assuming stationarity. Consistency results of the algorithm is provided.

**Strengths:**

1. Consistency results of the algorithm was shown even though the algorithm is relatively complicated.

2. The algorithm shows superior performance against the baselines.


**Weaknesses:**

1. The motivation for generalizing PCMCI to handle semi-stationary processes is not clear, especially from a theoretical perspective. The description of PCMCI is completely moved to the supplementary. I think including a brief description and the key procedures (e.g., how the superset is estimated) is helpful for understanding why PCMCI is good fit for the considered setting.

2. As mentioned in Section F. from Appendix, the proposed algorithm does not yield good performance in finite samples until the heuristic method $turning\text{ } points$ is introduced. To support the theoretical results, it should be shown that the algorithm starts to work when the sample size is sufficiently large.

typo:
Definition 2.3 is not correct: $\Pi_{k}^{j}(T)$ depends on $k$, but its definition in (8) does not depend on $k$.



**Questions:**

Since PCMCI assumes stationarity, in Algorithm 1, why PCMCI is used to estimate the superset $\widehat{SPa}(X_{t}^{j})$ for semi-stationary processes?

**Limitations:**

No potential negative societal impact.

---

> ### Author Rebuttal · Authors · 2023-08-10
>
> We really appreciate the feedback. Regarding your questions,
>
> ## Q1. Description of PCMCI
>
> Thank you for the suggestion. In the final version of the paper, we will provide a concise overview of how PCMCI functions, utilizing the additional page available in the camera-ready version.
>
> Here is a brief overview of PCMCI. There are two stages of PCMCI: the condition-selection stage and the causal discovery stage. In the first stage, unnecessary edges are removed based on the conditional independencies from an initialized partially connected graph. In the second stage, Momentary Conditional Independence tests (MCI) are used to further remove the false positive edges caused by autocorrelations in data.
>
> ## Q2. Relevance of PCMCI
>
> After running PCMCI on the whole multi-variate time series with Semi-Stationary SCM, the parent sets  $\widehat{SPA}(X^{j}_{t}),j\in [n]$ we obtained (line 2 in **Algorithm 1**) for each variable should be a superset of its true parent set for any time point $t\in T$, proved by **Lemma 3.2-3.3**.
>
> For a guess of $\hat{\omega}_{j}$, we can construct a corresponding series of time partition subsets. A series of MCI tests are conducted on samples whose time points are from the same time partition subset.
>
> Based on the result of the MCI tests, variables in $\widehat{SPA}(X^{j}_{t})$ will be removed based on the conditional independencies.
>
> If $\hat\omega_{j}=\omega_{j}$, the samples we used in each MCI tests are from the same causal mechanism and $\widehat{SPA}(X^{j}_{t})$ will shrink to the true parent set under assumption **A1-A7** with an oracle (infinite sample size limit).
>
> If $\hat\omega_{j}\neq\omega_{j}$, however, the samples in MCI tests are from different causal mechanisms, bringing in more causal relations under assumption  **A6**. Hence according to the MCI test results, $\widehat{SPA}(X^{j}_{t})$ will shrink (or not shrink) to a wrong parent set whose size is larger than the true parent set, leading to a denser causal graph, proved by **Lemma 3.4**.
>
> Hence, guess $\hat \omega_{j}$ resulting in the most sparse causal graph equal to the correct estimation $ \omega_{j}$. Based on the consistency of PCMCI algorithm and our proof of **Lemma 3.2-3.4**, the proposed algorithm PCMCI$_\Omega$ is sound under assumption **A1-A7** with an oracle (infinite sample size limit).
>
> ## Q3. PCMCI$_\Omega$ without *turning point*
> Thank you for this insightful comment, and we will add the related discussion in the camera-ready paper.
>
> We have conducted new experiments without using *turning point* and therefore $\hat\omega_{j}$ has been chosen according to the original rule stated in line 12 in **Algorithm 1**. The results show that algorithm shows similar performance with and without the *turning point*.
>
> Shown as **Fig.1(a)** in the Rebuttal pdf file, PCMCI$_\Omega$ without *turning point* results in a slightly larger standard error with a smaller sample size.
>
> As time length $T$ increases, the performance of the algorithm without *turning point* has consistently increased and is even slightly better than PCMCI$_\Omega$ with *turning point*.
>
> The consistent performance of PCMCI$_\Omega$ under different chosen rules of $\omega$ supports our theoretical result, that is, the correct periodicity $\omega$ leads to the most sparse causal graph.
>
> ## Q4. Updated Definition 2.3 *Time Partition*
> Thank you for pointing this out. This definition in the main paper is not precise. The updated definition should be as follows.
>
> A time partition $\Pi^{j}(T)$ of a univariate time series $X^{j}$ in a Semi-Stationary SCM with periodicity $\omega_{j}$ is a way of dividing all time points $t\in[T]$ into a collection of non-overlapping non-empty subsets $\Pi^{j}_{k} (T),k\in [\omega_j]$ such that
>
> $\Pi^j_k(T):=\\{t:\tau_{\max}+1 \leq t \leq T, (t \bmod \omega_j)+1=k \\} $
>
> where $\bmod$ denotes the modulo operation. For instance, $5\bmod3=2$.

---

> > ### Comment · Reviewer_nsZ6 · 2023-08-15
> > **Reply to the rebuttle**
> >
> > Q3: It is good to see that the algorithm does not heavily rely on turning points. This addresses my major concern of the algorithm. Based on this, I would raise my score slightly.
> >
> > Q1: I am still quite skeptical about using PCMCI for estimating $\hat{SPa}(X_{t}^{j})$ in finite samples, since PCMCI is proposed for stationary processes rather than semi-stationary processes. The proofs of Lemma 3.1 and 3.2 assume consistent CI tests, so I think the theoretical guarantees are quite limited. But I get that it is quite challenging to obtain stronger results.
> >
> > Overall, I think the studied setting is interesting and method has some novelty. But since semi-stationary processes defined in this paper are not studied in the literature, it is not clear whether it is general enough to model real world periodic random processes. Therefore, experiments on real data with ground truth will make the method more convincing.

---

> > > ### Author Response · Authors · 2023-08-15
> > >
> > > **Q3**: Thank you for the reconsideration.
> > >
> > > **Q1**: Thank you for these valuable comments. The CI tests in both PCMCI and our algorithm are assumed to be consistent given i.i.d. samples. We do not assume the consistency of CI tests with respect to semi-stationary data. Therefore any CI tests that maintain consistency with i.i.d. samples can be seamlessly integrated into our algorithm. This applies not only during the initial PCMCI phase but also in the subsequent step of our algorithm, where Momentary Conditional Independence (MCI) tests come into play. For instance, CI tests with proven consistency, such as Chi-square tests or Fisher's exact tests, can be employed in both PCMCI and our algorithm.
> > >
> > > This is feasible because, in both cases, variables are conditioned on parent sets or supersets of parent sets, ensuring the independence of samples due to the independence of exogenous noise terms. By choosing a correct $\omega$, conditioning on a superset of the parent set can yield identical samples from the same causal mechanism. An incorrect $\omega$ results in samples from a mixture distribution, as outlined in Eq.50 within the supplementary material. Hence it may introduce more dependent relationships in the mixture distribution, supported by **Lemma 3.2**.
> > >
> > > Once again, we express our heartfelt appreciation for your insightful feedback. We aspire that our algorithm serves as an innovative tool, harnessing expertise from various domains to unveil the latent periodic patterns within causal mechanisms across real-world situations.

---

### Author Rebuttal · Authors · 2023-08-10

Dear reviewers,

We extend our sincere gratitude for dedicating your time to meticulously assess our submission and offer invaluable insights.

With the initial rebuttal phase comes to an end, we believe that our response effectively addresses the raised points of concern.

We have attached a one-page pdf with new experiment results.

We remain fully open to continued discussions throughout the forthcoming discussion phase.

In light of our responses and the implemented revisions, we kindly request you to reconsider your evaluation.

Warm regards, Authors

---

### Decision · Program_Chairs · 2023-09-21

**Decision:**

Accept (poster)

**Comment:**

This paper is concerned with causal discovery method for semi-stationary time series, where causal structure may periodically change over time. A constraint-based, non-parametric, structural learning algorithm is proposed based on the PCMCI approach. It is extended and adapted to cope with the considered semi-stationary setting. The causal discovery problem is tackled by standard conditional independence tests. The paper shows that the proposed algorithm is sound in the discovered causal relationships. Experiments on both synthetic and real-world data are reported to illustrate the good performance of the proposed approach. Overall, the considered setting is practical and theoretical and empirical results in the paper demonstrate that the proposed method is a nice contribution to the community.